# Confined-microtubule assembly shapes three-dimensional cell wall structures in xylem vessels

Takema Sasaki [1], Kei Saito [2,3], Daisuke Inoue[4], Henrik Serk [5], Yuki Sugiyama [1,6], Edouard Pesquet [5,7], Yuta Shimamoto [2,3] & Yoshihisa Oda [1] ✉

Properly patterned deposition of cell wall polymers is prerequisite for the morphogenesis of plant cells. A cortical microtubule array guides the two-dimensional pattern of cell wall deposition. Yet, the mechanism underlying the three-dimensional patterning of cell wall deposition is poorly understood. In metaxylem vessels, cell wall arches are formed over numerous pit membranes, forming highly organized three-dimensional cell wall structures. Here, we show that the microtubule-associated proteins, MAP70-5 and MAP70-1, regulate arch development. The *map70-1 map70-5* plants formed oblique arches in an abnormal orientation in pits. Microtubules fit the aperture of developing arches in wild-type cells, whereas microtubules in *map70-1 map70-5* cells extended over the boundaries of pit arches. MAP70 caused the bending and bundling of microtubules. These results suggest that MAP70 confines microtubules within the pit apertures by altering the physical properties of microtubules, thereby directing the growth of pit arches in the proper orientation. This study provides clues to understanding how plants develop three-dimensional structure of cell walls.

Xylem vessels in vascular plants interconnect the entire plant body and distribute water and minerals from their roots to leaves. Xylem vessel elements deposit secondary cell walls in distinct patterns, adjusting the mechanical strength of the vessels to avoid their collapse under the negative pressure caused by the water transport[1,2]. The protoxylem vessel elements organize their secondary cell walls in annular or spiral patterns, while the metaxylem vessel elements organize them in reticulate or pitted patterns. These distinct patterns are defined by the specific arrangement of the cortical microtubule array due to the action of specific microtubule associated proteins (MAPs) in each type of xylem vessels[3–7]. In the Arabidopsis protoxylem cells, the arrangement of the cortical microtubules into the annular/spiral patterns is facilitated by several proteins including the cellulose synthase- and microtubule-interacting protein CSI/POM1, microtubule severing protein KTN1, and the microtubule nucleation complexes[8–10]. In metaxylem cells, cortical microtubules are locally disassembled by the MIDD1-Kinesin-13A complex to form pits of secondary cell walls[11,12], directed by the local activation of the ROP G proteins[13,14]. CORD and IQD13 regulate the cortical microtubule array to adjust the planar shape of secondary cell wall pits[15,16]. In *Brachypodium distachyon*, microtubule associated protein MAP20 regulates the size of secondary cell wall pits in metaxylem vessels[17].

Besides the above-mentioned patterns of secondary cell walls, three-dimensional cell wall arches have evolved that maintain the pit

[1]Department of Biological Science, Graduate School of Science, Nagoya University, Furo-cho, Chikusa-ku, Nagoya, Aichi, Japan. [2]Department of Chromosome Science, National Institute of Genetics, Mishima, Shizuoka, Japan. [3]Department of Genetics, SOKENDAI University, Mishima, Shizuoka, Japan. [4]Factuly of Design, Kyusyu University, Fukuoka, Japan. [5]Umeå Plant Science Centre (UPSC), Department of Plant Physiology, Umeå University, Umeå, Sweden. [6]Institute for Advanced Research, Nagoya University, Furo-cho, Chikusa-ku, Nagoya, Aichi, Japan. [7]Arrhenius laboratories, Department of Ecology, Environment and Plant Sciences (DEEP), Stockholm University, Stockholm, Sweden. ✉e-mail: oda.yoshihisa.w5@f.mail.nagoya-u.ac.jp

strength and water permeability while reducing their vulnerability to cavitation and embolism[5,18]. The cell wall arches arise from the boundaries of pits and grow over the cytoplasmic side of the pit membranes, areas where cell walls are not thickened nor lignified (Fig. 1a). The BDR-WAL complex stabilizes actin microfilaments at the boundary of pits to promote the formation of the cell wall arches[19]. Yet, the detailed mechanisms underlying the arch growth is poorly

understood. It is particularly unknown how microtubules are organized within pits to develop the proper structure of cell wall arches.

MAP70 is the plant-unique MAP family[20]. The two interacting paralogs of the MAP70 family, the MAP70-5 and MAP70-1, localize to the boundary of secondary cell walls in the Arabidopsis cultured xylem cells[21]. Overexpression and knockdown of MAP70-1/MAP70-5 modulated the deposition patterns of secondary cell walls in the cultured

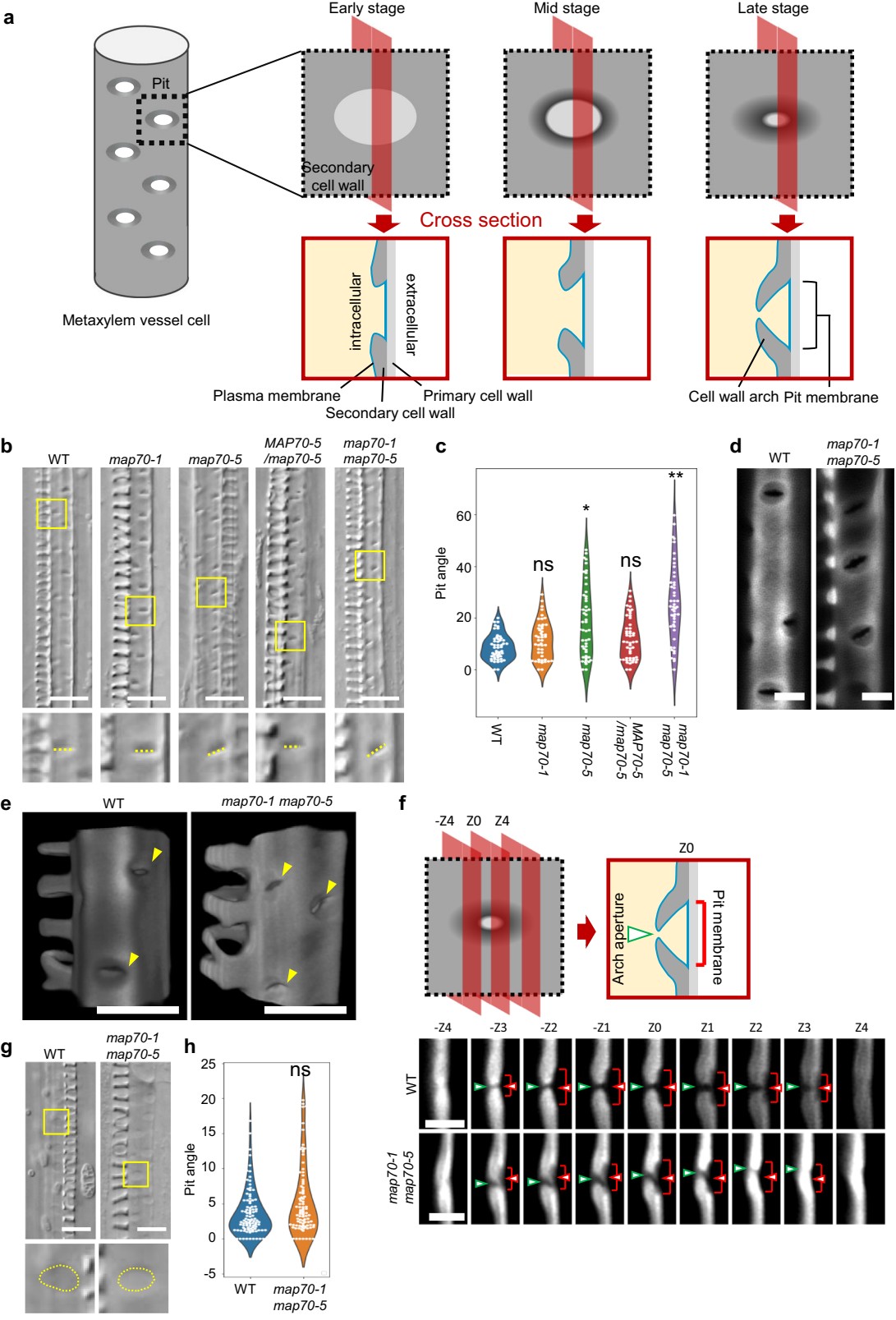

**Fig. 1 | MAP70 regulates the angle of secondary cell wall pits. a** Schematic of pit formation in metaxylem cell. **b** Differential interference contrast (DIC) images of xylem vessels in roots of wild type (WT), *map70-1*, *map70-5*, *map70-1 map70-5* (*map70W*), and *map70-5* expressing *pMAP70-5:MAP70-5-GFP* (*MAP70-5/map70-5*). Yellow boxes indicate the magnified images below. **c** Angle of secondary cell wall pits. *$p < 0.05$; **$p < 0.01$; ns, not significant; ANOVA with Tukey's honest significant difference ($n = 50$ pits). **d** Secondary cell wall pits of wild type and *map70-1 map70-5*. Cell walls were stained with Basic Fuchsin. The images are representative of five independent experiments. **e** Volume rendering of secondary cell walls. Yellow arrowheads indicate pits. **f** Cross-sections of secondary cell wall pits with 0.4 μm steps through the pit area from -Z4 to +Z4 positions. Cell walls were stained with Basic Fuchsin. The images are representative of five independent experiments. **g** DIC images of xylem vessels at the early stage of differentiation. Yellow boxes indicate the magnified images below. (h) Angle of secondary cell wall pits. Ns, not significant; Student's *t*-test ($n = 100$ pits). Scale bars: 10 μm **b**, **g**; 5 μm **d**, **e**; 2 μm **f**.

xylem cells[21]. The subcellular localization of MAP70-1/MAP70-5 led us to speculate their possible involvement in the BDR-WAL-dependent arch formation and in the microtubule organization within pits.

In this study, we investigated whether MAP70-5/MAP70-1 is involved in pit formation and the BDR-WAL pathway in Arabidopsis. We show that *MAP70-5* and *MAP70-1* are required for directing the formation of pit arches in the proper orientation. MAP70-5 localized to the aperture of pit arches independently of BDR-WAL. Loss of *MAP70-5* and *MAP70-1* causes the abnormal extension of microtubules beyond the pit boundaries and the formation of tilted pit arches. In vitro microtubule assays revealed that MAP70-5 crosslinks microtubules and reduces their stiffness. These activities of MAP70-5 promoted bundling and bending of microtubules, resulting in the formation of microtubule rings. Together, our results indicate that MAP70 confines microtubules within the pit aperture by altering the physical property of microtubules, thereby directs the growth of pit arches in the proper orientation.

## Results

### MAP70-5 and MAP70-1 regulate the direction of pits

To understand the functions of *MAP70-5* and *MAP70-1* in pit formation, we introduced a mutation into the *MAP70-5* and its interacting paralog *MAP70-1* using CRISPR/Cas9 technology. We isolated plants harboring an adenine insertion in the second exon of *MAP70-1*, and a 43 base-pair (bp) deletion at the third exon-intron junction of *MAP70-5* (Supplementary Fig. 1). The mRNA sequence revealed that the *map70-1* mutation caused a codon shift, and the *map70-5* mutation caused a translation of the second intron, both resulting in a premature stop codon (Supplementary Fig. 1), suggesting that they are null. The growth of *map70-1 map70-5* plants was comparable with that of the wild type plants (Supplementary Fig. 2). However, in the *map70-5* mutant, pits of metaxylem vessels in roots were oblique ~10–20° to the transverse axis of the vessel, whereas transverse pits were formed in wild type cells (Fig. 1b, c). The addition of *map70-1* mutation enhanced the *map70-5* phenotype tilting even more pits off the transverse axis to ~30° (Fig. 1b, c). Genetic complementation of *map70-5* with *pMAP70-5:MAP70-5-GFP* were fully restored pits to transverse orientation (Fig. 1b, c), demonstrating that *MAP70-5* is responsible for the pit orientation. Confocal sectioning (Fig. 1d) and volume rendering (Fig. 1e) moreover revealed that the slit within the pits, which is the aperture of cell wall arches, seemed to be inclined in *map70-1 map70-5* cells.

Cell wall arches develop partly over the pit membrane. To clarify which structure in the pits was inclined in *map70*, we observed the relative positions of the pit aperture and pit membrane in the sequential cross sections (Fig. 1f). In wild type plants, the pit aperture was found just above the center of the pit membrane throughout the sections. By contrast, in the *map70-1 map70-5* mutant, the position of pit aperture was shifted from the center of the pit membrane (Fig. 1f), suggesting that the oblique pits of *map70* mutants was unlikely due to the angle of pit membrane but due to the abnormal structure of cell wall arches. Indeed, during the early stage of differentiation, when arches had not yet developed, the *map70-1 map70-5* mutant had transverse pits that were comparable with those in wild type plants (Fig. 1g, h). These results suggest that the *map70* mutants form normal pit membranes but thereafter develop cell wall arches in oblique orientation, resulting in the formation of the tilted pit apertures.

### MAP70-5 localized to microtubules at the tips of arches

Consistent with the tilted pit phenotype of *map70-5* plants, a glucuronidase (GUS) reporter of the 2 kbp *MAP70-5* promoter showed that *MAP70-5* was expressed in the xylem tissues (Supplementary Fig. 3). To further investigate how MAP70 regulates the structure of cell wall arches, we observed MAP70-5 and microtubules, labeled with *pMAP70-5:MAP70-5-tagRFP* and *pIRX3* (*IRREGULAR XYLEM 3* promoter)*:EYFP-TUB6*, respectively, in *map70-1 map70-5* roots. MAP70-5 preferentially colocalized with microtubules in the pits of metaxylem (Fig. 2a), although some MAP70-5-tagRFP signals were faintly observed outside the pit region. The tagRFP/EYFP intensity ratio in pits was higher than that outside pits, demonstrating the preferential localization of MAP70-5 at the microtubules in pits (Fig. 2b). In contrast to metaxylem cells, MAP70-5-GFP in the protoxylem was localized beneath the cell wall bands labeled with propidium iodide (PI) (Fig. 2c).

We then evaluated the localization of MAP70-5 compared to PI-stained secondary cell walls from the early to late stages of metaxylem cell differentiation in roots. During the early stage of differentiation, MAP70-5-GFP was almost absent in the pits (Fig. 2d, top). However, once the nascent arches emerged, MAP70-5-GFP appeared at the tips of the arches, where they persisted as the arches grew (Fig. 2d, 2nd to 4th panels from the top). These observations indicated that MAP70-5 is specifically localized at the tip of growing pit arches.

### MAP70 is spatially separated from BDR-WAL and ROP-MIDD1

We have previously shown that at the pit boundaries the ROP interactor BDR1 recruits the actin-binding protein WAL to promote the formation of cell wall arches[19]. To clarify the relationship between these protein complexes, we observed MAP70-5-GFP and BDR1-tagRFP in root xylem vessels. During cell wall arch growth, BDR1-tagRFP remained at the base of the arches, whereas MAP70-5-GFP localized at the tips of the cell wall arches (Fig. 2e). During the early stages of arch formation, the MAP70-5-GFP signal was not uniform but was sparsely distributed as puncta around the pit boundaries where pit arches were emerging. As the further arches developed, MAP70-5-GFP gradually accumulated at both ends of the arches (Fig. 2e, white box). Projection of the MAP70-5 GFP signals in all five pits examined demonstrated the preferential localization of MAP70-5 at the ends of pit arches (Fig. 2f).

On the pit membranes, activated ROP GTPases promote microtubule depolymerization by recruiting the MIDD1-Kinesin-13A complex[11,13,19]. To clarify the spatial relationship between MAP70-5 and ROP, *pMIDD1:MIDD1ΔN-ECFP*, which marks the active form of ROP, was observed together with *pMAP70-5:MAP70-5-EYFP* and *pBDR1:BDR1-tagRFP*. The three markers were clearly spatially separated and confined to distinct areas of the pits (Fig. 2g). The localization pattern of MAP70-5-GFP and BDR1-GFP was not affected in *wal* and *map70-1 map70-5*, respectively (Fig. 2h, i). These observations indicate that MAP70-5 is spatially separated from BDR-WAL and ROP-MIDD1.

### MAP70 confines microtubules in pits

To determine the function of MAP70 at the tip of arches, we compared microtubule alignment at the tips of arches between wild type plants and *map70-1 map70-5* mutants. In both genotypes, microtubules were intensely observed at the boundaries of pits (Fig. 3a). However, when

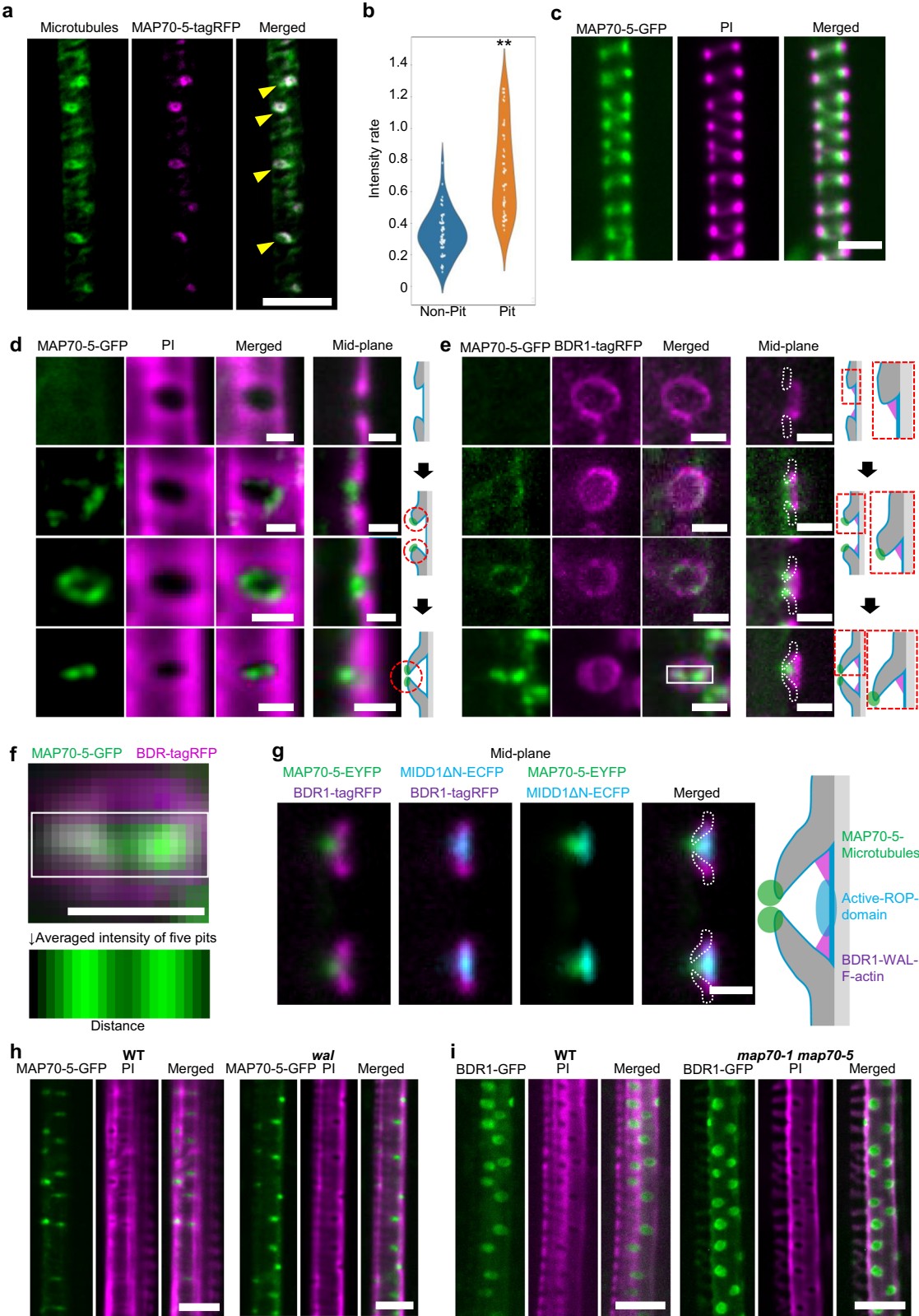

cell walls were stained with PI, we noticed that the alignment of the microtubules at the edge of pits were different between wild type and the mutant plants. In wild type plants, microtubules were confined to the area of pits (Fig. 3b). By contrast, the microtubules in *map70-1 map70-5* mutants extended beyond the boundaries of the pit edges (Fig. 3c). The ratio of microtubule length to the pit length in the

*map70-1 map70-5* cells was 1.2-fold larger than that of wild type cells (Fig. 3d), confirming our observation.

To reveal the behavior of microtubules at pit arches, we observed the microtubule end-tracking protein EB1b, which forms comet-like streaks on growing microtubule plus-ends. Comets of GFP-EB1b signals were detected broadly at the cell cortex in wild type

**Fig. 2 | MAP70-5 localizes to the tip of the cell wall arch in metaxylem vessel cells. a** Microtubules (*pIRX3:EYFP-TUB6*) and MAP70-5-tagRFP (*pMAP70-5:MAP70-5-tagRFP*) in metaxylem vessel cells. Yellow arrowheads indicate pits. The images are representative of five independent experiments. **b** Intensity of MAP70-5-tagRFP signals relative to EYFP-TUB6 at the pit area (Pit) and the non-pit area (Non-pit). **p < 0.01; Student's *t*-test (*n* = 50 areas). **c** MAP70-5-GFP (*pMAP70-5:MAP70-5-GFP*) in protoxylem vessel cells in *map70-5*. Cell walls are labeled with propidium iodide (PI). The images are representative of five independent experiments. **d** MAP70-5-GFP (*pMAP70-5:MAP70-5-GFP*) in metaxylem vessel cells of *map70-5*. From top, early, middle, late, and mature stages of differentiation. Cell walls are labeled with PI. The images are representative of five independent experiments. **e** MAP-70-5-GFP (*pMAP70-5:MAP70-5-GFP*) and BDR1-tagRFP (*pBDR1:BDR1-tagRFP*) in metaxylem vessel cells of *map70-5*. From top, early, middle, late, and mature stages of differentiation. The images are representative of five independent experiments. **f** Vertical average intensity of MAP70-5-GFP of the area shown in the white box of E. Data are the mean of five pits. **g** MAP70-5-EYFP, active ROP (MIDD1ΔN-ECFP), and BDR1-tagRFP at pits in metaxylem vessel cells of wild type plants. The images are representative of five independent experiments. **h** MAP70-5-GFP in metaxylem vessel cells of wild type (WT) and *wal*. Cell walls are labeled with PI. The images are representative of five independent experiments. **i** BDR1-GFP in metaxylem vessel cells of *map70-1 map70-5*. The images are representative of five independent experiments. Scale bars: 10 μm **a, c, h, i**; 2 μm **d, e, g**.

and *map70-1 map70-5* metaxylem cells (Fig. 3e; Supplementary Movie 1, 2). However, the projection of the time-lapse images highlighted the distinct trajectories of EB1b, which were most likely the microtubules at the tip of growing pit arches (Fig. 3f). In the wild type cells, the EB1b trajectory formed small circles (Fig. 3f), probably representing the pit boundaries. By contrast, in *map70-1 map70-5* cells, the EB1b trajectories formed short rods with extended tails, consistent with the microtubules of the mutant pits (Fig. 3f). When we focus on the single comets of the EB1b, in the wild type cells, the EB1b comets moved along the arc of the circular trajectory of EB1b (Fig. 3g). In the *map70-1 map70-5* cells, however, the comets moved outwards along the tail of the EB1b trajectory (Fig. 3g). These results suggest that at least some of microtubules in the pits are highly dynamic and that MAP70 prevents the dynamic microtubules from growing beyond the pit arches. In addition, we noticed that the orientation angle of pits was correlated to the alignment of cortical microtubules outside pits in *map70-1 map70-5* plants. The cortical microtubules were aligned at 0–30° in both the wild type plants and *map70-1 map70-5* mutants (Fig. 3h). In wild type plants, the orientation angle of the pits was maintained at 0-10° regardless of the orientation angle of the cortical microtubules. By contrast, in the *map70-1 map70-5* mutants, the orientation angle of the pits was tilted to the angle correlated with the orientation angle of the cortical microtubules (Fig. 3i). Considering that MAP70-5 localized specifically to cell wall arches, it is plausible that MAP70-5 confines microtubules within the tips of the cell wall arches, thereby uncoupling the pit orientation from the surrounding cortical microtubules.

## MAP70 promotes bending of microtubules in vivo

To understand how MAP70 confines microtubules to pit arches, we examined the effects of transient *MAP70-5-GFP* expression on the microtubule array in leaf epidermal cells of *Nicotiana benthamiana*. *MAP70-5-GFP* was introduced to the leaf epidermis under the control of the *LexA* estrogen-inducible promoter, together with the *pUBQ10:tagRFP-TUB6* marker. Surprisingly, after treatment with estrogen, the cortical microtubules bended, gradually assembled, and finally formed circular bundles (Fig. 4a, b; Supplementary Movie 3). Similar behaviors of microtubules were also observed in the epidermis of leaves and hypocotyls, and root hairs of the transgenic Arabidopsis plants in which GFP-MAP70-5 was ectopically expressed (Supplementary Fig. 4).

MAP70-5 has four predicted coiled-coil domains[20] (Fig. 4c). To determine which domains were responsible for microtubule bending, a deletion series of MAP70-5 was fused with GFP and expressed in the leaf epidermis. MAP70-5 lacking the third and fourth coiled-coiled domains (MAP70-5ΔCC34) localized to the microtubules and caused the microtubules to bend (Fig. 4d, e). MAP70-5 lacking the first and second coiled-coil domains (MAP70-5ΔCC12) localized to the microtubules but did not cause the microtubules to bend (Fig. 4d, e). These results indicate that the N-terminal half of MAP70-5 possesses the microtubule-bending activity, and that MAP70-5 has at least two sites that can interact with microtubules directly or indirectly, one in the N-terminal half and the other one in the C-terminal half.

Next, we examined the function of the first and second coiled-coil domains. The deletion of the first coiled-coil domain (MAP70-5ΔCC1) did not affect microtubule curvature. By contrast, partial deletions of the second coiled-coil domain, MAP70-5ΔCC2a, MAP70-5ΔCC2b, and MAP70-5ΔCC2c, reduced microtubule curvature to the level of the control, although they still localized to microtubules (Fig. 4d, e). This result indicated that the entire second coiled-coil domain is required for the microtubule-bending activity of MAP70-5.

Finally, we tested whether the microtubule bending activity of MAP70-5 is required for arch formation by introducing the MAP70-5ΔCC2 series to *map70-1 map70-5* plants. Similarly, to the full length of MAP70-5, MAP70-5ΔCC2c was localized to pits of the *map70-1 map70-5* mutant (Fig. 4h). However, none of the MAP70-5ΔCC2 deletion series rescued the tilted orientation of pits of the *map70-1 map70-5* mutants (Fig. 4g). These results indicate that the second coiled-coil domain of MAP70-5, which possesses the microtubule-bending activity, is not required for their localization to pits but is necessary for the regulation of pit orientation.

## MAP70 promotes cross-linking and increases flexibility of microtubules in vitro

To understand the mechanistic activity of MAP70-5 on microtubule behavior, we produced recombinant MAP70-5 (6×His-GFP-MAP70-5) and performed in vitro co-incubation assays with fluorescently-labeled microtubules. First, we investigated the binding behavior of MAP70-5 on microtubules using in vitro single-molecule assays. Biotinylated X-rhodamine-microtubules were immobilized onto the surface of neutravidin-coated chambers. A low (3 nM) concentration of recombinant GFP-MAP70-5 was then perfused into the chamber, and the sample behavior was imaged under total internal reflection fluorescence (TIRF) microscopy. GFP-MAP70-5 signals were observed along microtubules (Fig. 5a). The signals moved along the length of the microtubules in both directions (Fig. 5b; Supplementary Fig. 5; Supplementary Movie 4). The behavior of MAP70-5 suggested that MAP70-5 interacts not statically but diffusively with microtubule lattices.

As MAP70-5 was localized to the edge of pit arches, where microtubules were most steeply curved, we hypothesized that MAP70-5 preferentially binds to curved microtubules. To test this, we analyzed the positions of MAP70-5 and the curvature of microtubules. A high concentration (50 nM) of recombinant GFP-MAP70-5 was perfused into the chamber, and was rinsed away with buffer, leaving a fraction of GFP-MAP70-5 on X-rhodamine-microtubules (Fig. 5c). We then measured the intensity of GFP-MAP70-5 signals depending on its position on curved microtubules (Fig. 5d). No significant correlation could be observed between the intensity and microtubule curvature, suggesting that MAP70-5 did not respond to microtubule curvature for their binding.

During the single-molecule assays, we occasionally found that X-rhodamine microtubules were bundled by addition of recombinant GFP-MAP70-5. Indeed, recombinant GFP-MAP70-5 promoted formation of bundled microtubules in a concentration-dependent manner (Fig. 5e), implying the possibility that MAP70-5 has a microtubule

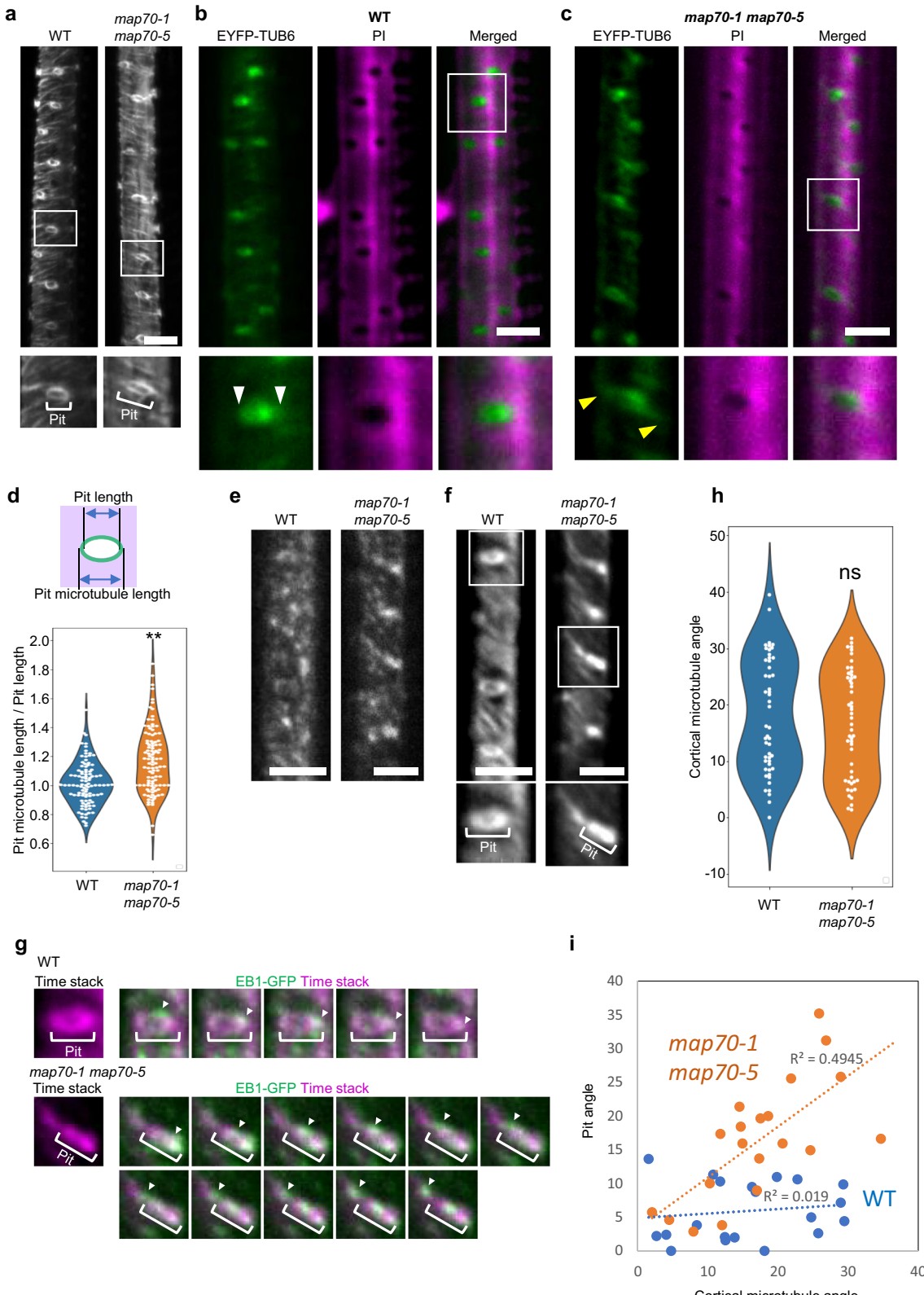

**Fig. 3 | MAP70-5 regulates microtubule arrays on pits. a** Cortical microtubules (*pIRX3-EYFP-TUB6*) in metaxylem vessel cells of wild type (WT) and *map70-1 map70-5*. The images are representative of five independent experiments. (b and c) Cortical microtubules (*pIRX3-EYFP-TUB6*) in metaxylem vessel cells of wild type **b** and *map70-1 map70-5* **c**, and magnified images of the areas outlined by white boxes. Cell walls are labeled with PI. Yellow arrowheads indicate microtubules extending beyond the edge of pits. **d** Ratio of length of pit microtubules and pits in wild type and *map70-1 map70-5*. **\*\*p < 0.01; Student's t-test (n = 100 pits). **e** EB1b (*pXCP1:GFP-EB1b*) in metaxylem vessel cells of wild type and *map70-1 map70-5*

plants. The images are representative of five independent experiments. **f** Projection of EB1b time-laps images. Time-lapse images were taken every 2 second. **g** Time-laps images of EB1b (green) around the secondary cell wall pits. The signals were overlayed with the projection of EB1b time-lapse images (magenta). **h** Angle of cortical microtubules in metaxylem vessel cells in wild type and *map70-1 map70-5*. ns, not significant; Student's *t*-test (n = 50 pits). **i** Correlation between the angle of pits and cortical microtubules around pits of wild type and *map70-1 map70-5* (n = 20 pits). Microtubule angle is the mean of two to five microtubules around each pit. Scale bars: 5 μm.

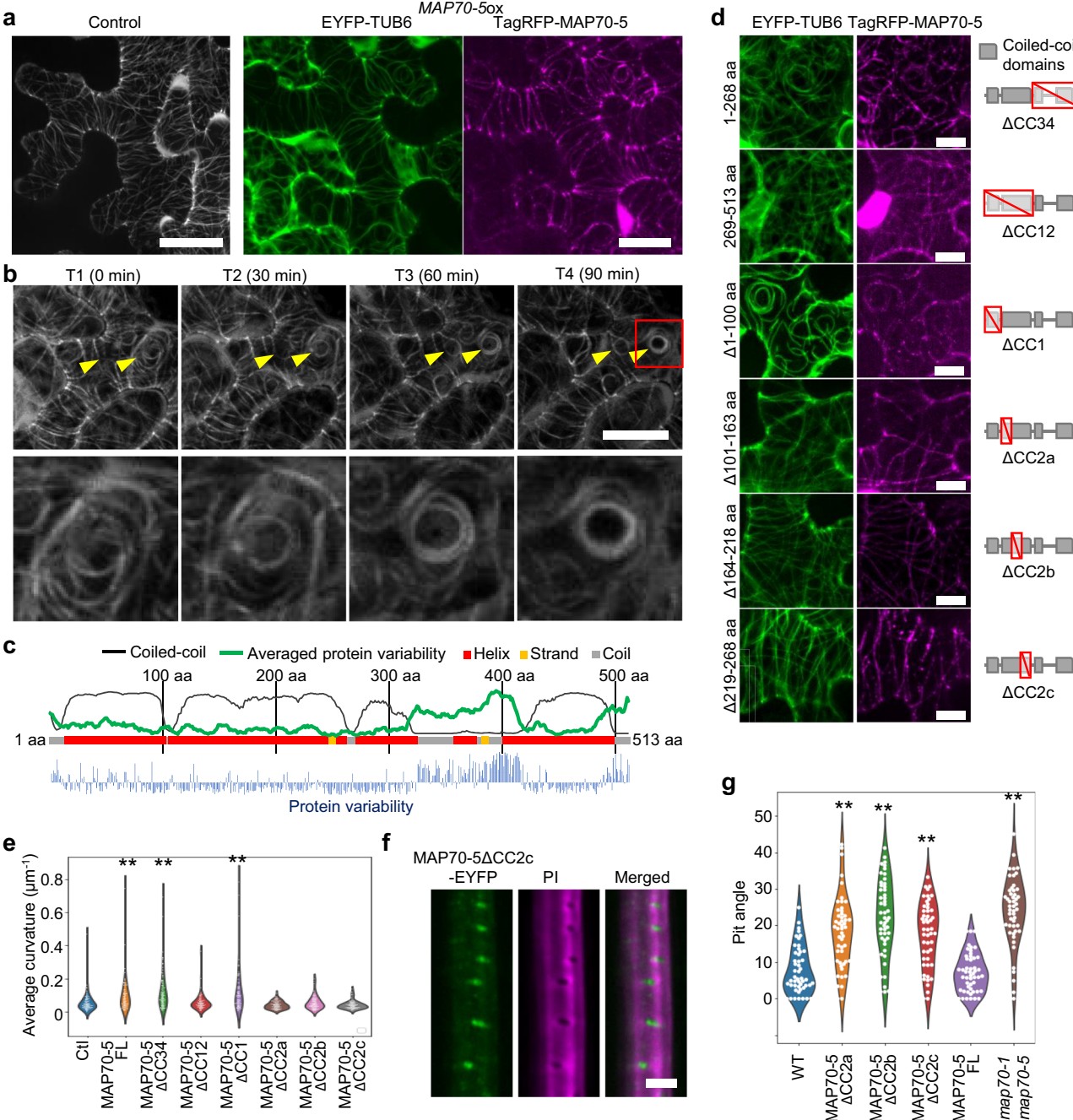

**Fig. 4 | MAP70-5 induces cortical microtubule bending. a** Cortical microtubules (*pUBQ10:EYFP-TUB6*) without (control) or with (right panels) tagRFP-MAP70-5 (*pLexA:tagRFP-MAP70-5*) in leaf epidermal cells of *N. benthamiana*. The images are representative of five independent experiments. **b** Time-lapse observation of cortical microtubules (*pUBQ10:tagRFP-TUB6*) in leaf epidermal cells of *N. benthamiana* expressing GFP-MAP70-5 (*pLexA:GFP-MAP70-5*). Yellow arrowheads indicate circular bundled microtubules. Red box indicates magnified images below. The images are representative of five independent experiments. **c** Predicted secondary structure of MAP70-5. Protein variability was calculated based on MAP70 families of land plants. **d** Truncated MAP70-5 (*pLexA:tagRFP-MAP70-5*) and cortical microtubules (*pUBQ10:EYFP-TUB6*) in leaf epidermal cells of *N. benthamiana*. **e** Average curvatures of cortical microtubules in leaf epidermal cells of *N. benthamiana* expressing full length (Ctl) or truncated *MAP70-5* (*pLexA:tagRFP-MAP70-5*). **f** Localization of truncated MAP70-5-EYFP (MAP70-5-ΔCC2c) in *map70-1 map70-5* metaxylem vessel cells. **g** Angle of secondary cell wall pits of wild type (WT), *map70-1 map70-5* expressing MAP70-5ΔCCa-c-EYFP or full-length MAP70-5-EYFP, and *map70-1 map70-5*. **\*\****p* < 0.01; ns, not significant; ANOVA with Tukey's honest significant difference (*n* > 50 pits). Scale bars: 20 μm **a**; 10 μm **b**, **d**; 5 μm **f**.

cross-linking activity. To clarify this possibility, we performed two-color microtubule cross-linking assay. Biotinylated and non-biotinylated microtubules, labeled with either X-rhodamine and Alexa Fluor 488 respectively, were incubated on a polyethylene glycol (PEG)-grafted, neutravidin-coated glass in the presence or absence of MAP70-5, and then rinsed with buffer. Biotinylated microtubules were able to directly bind to the glass by means of neutravidin, whereas non-biotinylated microtubules were rinsed away unless they were

crosslinked with the biotinylated microtubules via MAP70-5. In the absence of MAP70-5, non-biotinylated microtubules were rarely observed on the glass (Fig. 5f). By contrast, in the presence of MAP70-5, non-biotinylated microtubules remained on the glass, overlapping with the biotinylated microtubules (Fig. 5g). These results indicated that MAP70-5 possesses the capacity to crosslink microtubules.

Because MAP70-5 induced bending of microtubules in vivo, we hypothesized that MAP70-5 has an activity to influence the stiffness of

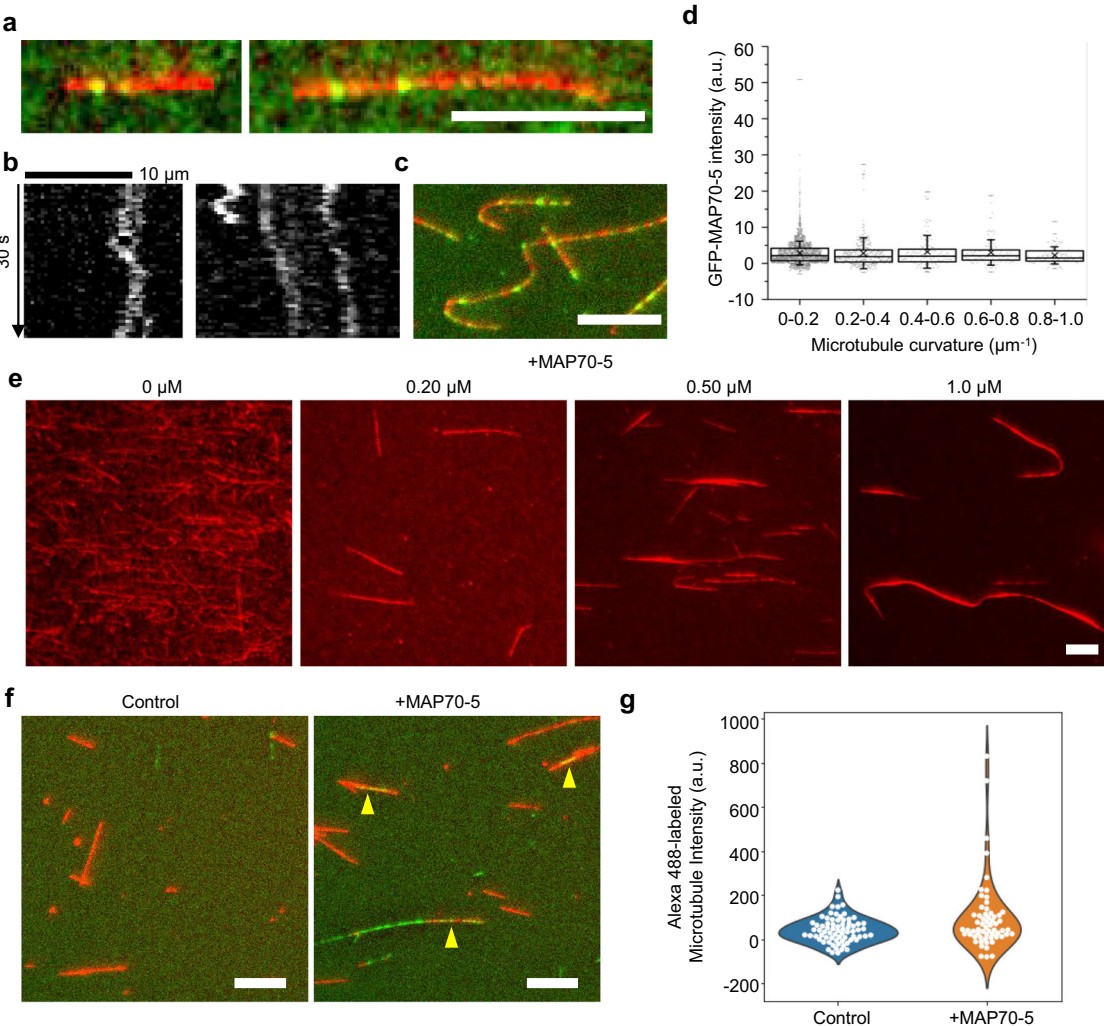

**Fig. 5 | MAP70-5 induces microtubule bundling in vitro. a**, **c** GFP-MAP70-5 (green) on X-rhodamine-microtubules (red). Microtubules were incubated with 3 nM **a** or 50 nM **c** 3 nM GFP-MAP70-5 followed by washout. **b** Diffusive movement of GFP-MAP70-5 on a microtubule. **d** Relationship between MAP70-5 intensity (GFP, at 50 nM) and microtubule curvature ($n = 6729, 430, 155, 119,$ and 125 for 0–0.2, 0.2–0.4, 0.4–0.6, 0.6–0.8, and 0.8–1.0 µm⁻¹, respectively). Center lines show the medians. Box limits indicate the 25th and 75th percentiles. Crosses, Whiskers, and a.u. indicate mean values, standard deviation, and arbitrary units, respectively. There was no significant correlation between GFP-MAP70 intensity and

microtubule curvature (one-way ANOVA, p = 0.15696). **e** Bundled microtubules (X-rhodamine-microtubules) after addition of MAP70-5 at 0–1.0 µM concentration. The images are representative of three independent experiments. **f** Microtubule crosslinking assay. Non-biotinylated Alexa Fluor 488-labeled microtubules (Green) and biotinylated X-rhodamine-microtubules (Red) in control buffer (Control) or with MAP70-5. Yellow arrowheads indicate overlap of non-biotinylated microtubules and biotinylated microtubules. **g** Intensity of colocalized Alexa 488-labeled microtubules with X-rhodamine-labeled microtubules in control buffer (Control, $N = 71$) and MAP70 buffer (+MAP70-5, $N = 61$). Scale bars: 10 µm **a**, **c**, **e**, **f**.

microtubules. Microtubules were tethered to the glass slide coated with the motor protein Kinesin-1 that enables microtubules to glide on the glass without affecting their physical flexibility on the glass slide. In the absence of MAP70-5, straight microtubules were gliding with slight curvatures on the glass (Fig. 6a; Supplementary Movie 5). By contrast, 10 sec after the addition of MAP70-5, microtubules started bending and glided along curved trajectories (Fig. 6a; Supplementary Movie 6). The persistence length of microtubules, which indicates the flexural stiffness of microtubules, was decreased by the addition of GFP-MAP70-5 (Fig. 6b)[22]. In addition to the bending, microtubules gradually formed bundles (Supplementary Fig. 6), and formed ring structures of ~2 µm in diameter (Fig. 6c, d), similar to the rings observed in tobacco leaves (Fig. 4b). These results suggest that MAP70-5 reduced the stiffness of microtubules, although cross-linking of microtubules via MAP70-5 might also contribute to such behavior of microtubules. To reduce the effect of the interaction between microtubules, we again performed the gliding assay at a low concentration of shorter microtubules (Fig. 6e; Supplementary Movie 7, 8). The time-lapse images of

the short microtubules were projected into single images, then the persistence length of the trajectories of the microtubules were quantified (Fig. 6f, g). In this condition, the persistence length of the microtubule trajectories was still decreased by the addition of MAP70-5 (Fig. 6g), supporting the possibility that MAP70-5 reduces the stiffness of microtubules. In this condition, we could also observe the process in which single microtubules form rings in the presence of MAP70-5 (Fig. 6h; Supplementary Movie 9).

To examine the effect of MAP70-5 exclusively on the stiffness of microtubules without interactions between microtubules, the micro beads were anchored to the flow chamber, then one end of microtubules was tethered to the beads, with the other end of the microtubules being free as a cantilever beam (Fig. 6i, j). In this condition, the tethered microtubules were slightly swaying in the flow chamber due to thermal fluctuation (Supplementary Movie 10). Then we could estimate the stiffness of the microtubules by measuring the curvature of the swaying microtubules. The persistence length of the tethered microtubules was decreased by the addition of MAP70-5 (Fig. 6k).

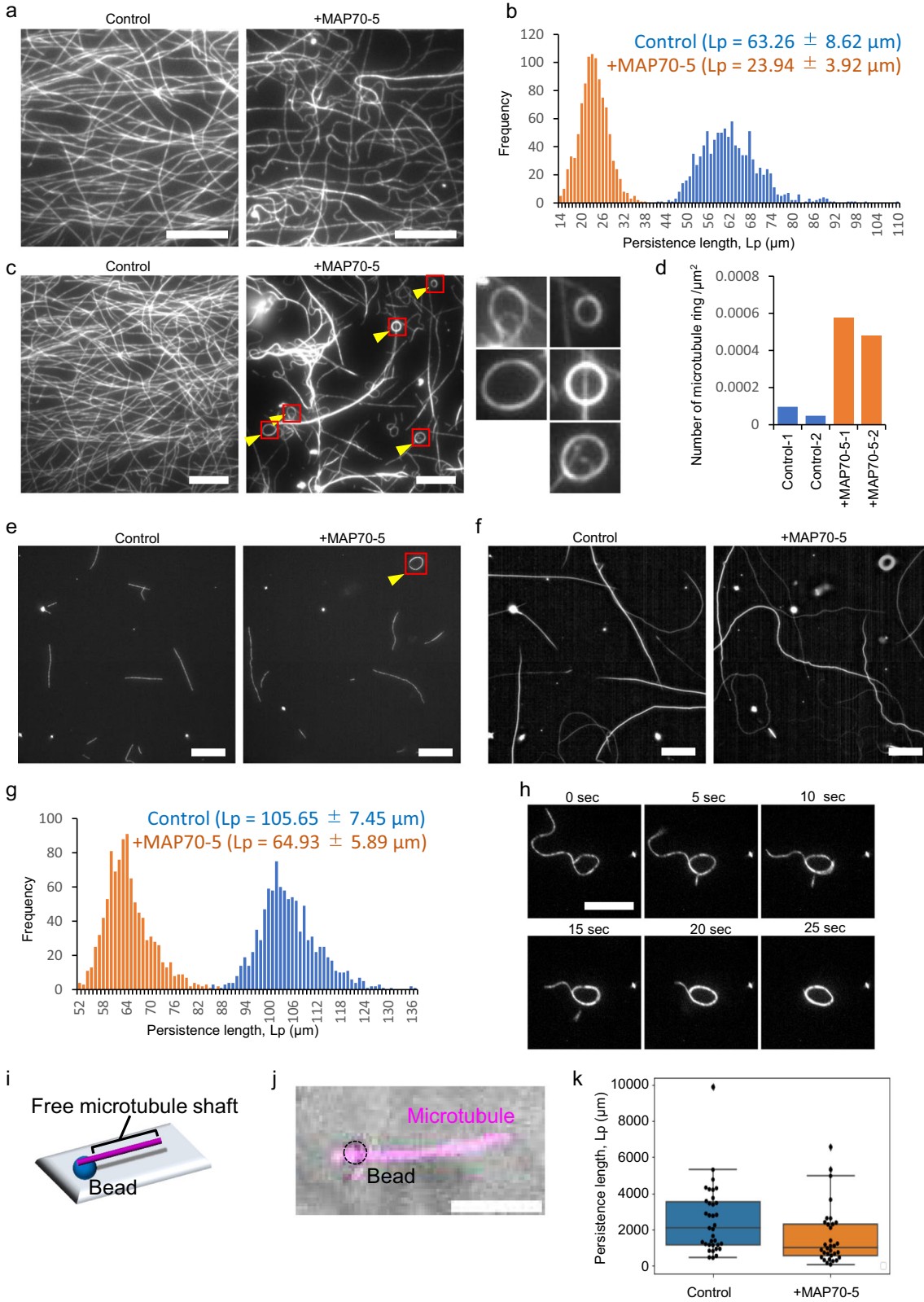

These results suggest that MAP70-5 reduces the stiffness of microtubules. Together with the results of the cross-linking and the gliding assays, our data suggest that MAP70-5 possesses two activities to increase the flexibility of microtubules and to promote bundling of microtubules. These two activities of MAP70-5 perhaps promoted microtubule bending when microtubules were propelled by the glass-anchored kinesins, eventually formed rings of microtubules, although we could not exclude the possibility that the MAP70-5 induces microtubule bending by leading a stabilized curvature in the microtubule polymer, not by increasing their flexibility.

## Discussion

In the herein study, we showed that MAP70-1 and MAP70-5 are required for the formation of arches in a transverse orientation.

**Fig. 6 | MAP70-5 induces microtubule bending and triggers ring formation of microtubules in vitro. a** ATTO565-microtubules in control buffer (Control) and 10 s after addition of 2 μM 6xHis-GFP-MAP70-5 ( + MAP70-5). The images are representative of two independent experiments. **b** Histogram of persistence length of microtubules 10 sec after addition of control buffer (Control) or GFP-MAP70-5-containing buffer (+MAP70-5). **c** ATTO565-microtubules 3 min after addition of control buffer (Control) or GFP-MAP70-5-containing buffer (+MAP70-5). Red box indicates magnified images. Yellow arrowheads indicate circular bundles of microtubules. The images are representative of two independent experiments. **d** Number of microtubule rings 3 min after addition of control buffer (Control) or MAP70-5 containing buffer (+MAP70-5). **e** Short ATTO565-microtubules in control buffer (Control) and 10 s after addition of 2 μM 6xHis- MAP70-5 ( + MAP70-5). The images are representative of two independent experiments. **f** Trajectory of gliding microtubules of **e**. Trajectory was created by projecting the time-lapse image stack. **g** Histogram of persistence length of trajectory of microtubules in control buffer (Control) or MAP70-5-containing buffer (+MAP70-5). **h** Process of ATTO565-microtubule ring formation. The images are representative of two independent experiments. **i** Schematic of microtubule one-side tethering assay. **j** Representative fluorescence images of a single microtubule with MAP70-5. **k** The persistence length of microtubules in control buffer ($n = 35$) or in the presence of 1 μM MAP70-5 ($n = 32$). Center lines indicate medians. Box plots indicate the 25th and 75th percentiles with S.D. as whiskers. Mean persistence length of microtubules in control and +MAP70-5 was 2525 ± 1873 μm (Median = 2071 μm) and 1603 ± 1576 μm (Median = 1020 μm), respectively. $p = 0.00679$ (Mann-Whitney U-test). Scale bars: 5 μm **j**, 10 μm **a**, **c**, **h**; 20 μm **e**, **f**.

MAP70-5 was specifically localized to the growing tip of cell wall arches and was capable of restricting the microtubules within pits. MAP70-5 had an activity to promote bending of microtubules in vivo. Our in vitro assays also showed that MAP70-5 had activities to promote crosslinking and bending of microtubules. The microtubule-bending activity of MAP70-5 was necessary for proper arch formation. It is therefore likely that the microtubule-bending activity of MAP70-5 allows microtubules to align along the tip of arches within pits. When such activity is absent, microtubules extend beyond the pit boundaries and are affected by the orientation of the cortical microtubule array, resulting in abnormal development of pits in an oblique direction. MAP70-5 is spatially independent of the ROP-MIDD1 and BDR-WAL complexes. Thus, at least three sets of regulatory complexes shape the three-dimensional structure of pits: (1) the ROP-MIDD1 defines the area of the pit membrane; (2) the BDR-WAL at the pit boundaries promotes the ingrowth of pit arches; and (3) the MAP70-5/MAP70-1 directs the growth of pit arches to the proper orientation. These three pathways should be spatially and temporally coordinated to form the fully functional structure of pits, which enables the lateral distribution of the water without reducing the hydraulic efficiency and resistance to embolism. An insightful future challenge will be investigating how these regulatory components are orchestrated within the tiny space of pits.

MAP70-5 is, to date, the only protein that is specifically associated with microtubules at the tips of arches in pits. The mechanism by which MAP70-5 localizes to the tips of arches remains unclear. Our results exclude the possibility that MAP70-5 responds to the microtubule curvature in pits. One possible mechanism might be that MAP70-5 senses the tension of the plasma membrane, which is stretched by the ingrowing tips of arches. Recently, supporting this possibility, it was reported that MAP70-5 was preferentially localized at the inner side of endodermal cells, which are mechanically forced out by the outgrowth of lateral root primordia[23].

One unique property clearly shown by our results is the capacity of MAP70-5 to promote the bending of microtubules. Another MAP, neuronal MAP6, is known to curve microtubules by incorporating into the lumen of microtubules during their polymerization[24]. We showed polymerized microtubules were bent by addition of MAP70-5, indicating that de novo polymerization of microtubules is not required for the bending by MAP70-5. This excludes the possibility that microtubules incorporate MAP70-5 into their lattice or lumen during polymerization like MAP6. Indeed, several MAPs and posttranslational modifications of tubulin have been found to alter the stiffness of microtubules[25–27], such as MAP65, which is capable of crosslinking microtubules[26]. It seems that the microtubule-bending by MAP70-5 relies on its capability of (1) reducing the stiffness of microtubules, (2) crosslinking microtubules, and (3) diffusive interaction with microtubules. The combination of these activities may induce the staggered assembly of the flexible microtubules with a steep curvature, resulting in the formation of the microtubule rings. We observed elongating microtubules around the pit boundary (Fig. 3g). Such microtubule behavior might facilitate remodeling of microtubule rings to form pit arches in transverse orientation.

Bending and curving of microtubules is widely found in plants. Bundling of the interphase cortical microtubules accompanies curvature of the interacting microtubules[28]. Microtubules need to bend at cell edges so that microtubules extend over steep geometries of cell walls[29]. Besides xylem pits, MAP70-5 is required for the remodeling of cortical microtubules during lateral root primordia formation[23]. MAP70-5 and its interacting paralogue, MAP70-1, are expressed not specifically in metaxylem cells[21]. Thus, interacting MAP70 paralogues might represent the versatile regulatory factors that facilitate the flexible organization of cortical microtubules in response to the geometry of cell walls by controlling microtubule stiffness.

## Methods
### Plant materials
*Arabidopsis thaliana* ecotype Columbia-0 was used as the wild type. Seedlings were grown on 1/2 Murashige and Skoog agar medium, containing 0.2% sucrose, or in soil at 22 °C under constant light.

### Constructs
The CRISPR/Cas9 vector for *MAP70* was generated as described previously[30,31]. A gRNA oligo with BbsI overhangs was generated by oligo annealing and cloned into the BbsI site of pEn-Chimera. The resulting clone was recombined with the pDe-Cas9 vector.

To generate *pMAP70-5:MAP70-5-GFP*, *pMAP70-5:MAP70-5-EYFP*, *pMAP70-5:MAP70-5-tagRFP*, and their truncated derivatives, a genomic fragment of the MAP70-5 coding sequence, including the 2 kbp promoter region, was PCR-amplified and cloned into the pENTR/D-TOPO vector (Thermo Fisher Scientific). Truncated derivatives were generated using inverted PCR. To fuse with GFP, EYFP, and tagRFP, the resulting clones were recombined with the pGWB504, pGWB540, and pGWB559 vectors[32], respectively, using LR clonase II (Thermo Fisher Scientific).

To generate *pMIDD1:ECFP-MIDD1ΔN*, the genomic fragment encoding amino acids 121 to 396 of *MIDD1* was PCR amplified and cloned into the pENTR/D-TOPO vector. The resulting clone was recombined with the pGWB445 vector[32]. The ECFP-MIDD1ΔN fragment was PCR amplified and cloned into the pENTR/D-TOPO vector. The 2 kbp promoter region of *MIDD1* was PCR amplified and inserted upstream of the ECFP-MIDD1ΔN start codon using an In-Fusion HD cloning kit (Takara). The resulting clone was recombined with the pGWB601 vector[33].

To generate *pLexA:GFP-MAP70-5*, *pLexA:tagRFP-MAP70-5*, and their truncated derivatives, coding sequences were PCR amplified and cloned into the pENTR/D-TOPO vector. Truncated derivatives were generated using inverted PCR. The resulting clones were fused with GFP or tagRFP, and inserted into the pER8 vector[34]. To generate 6×His-GFP-MAP70-5, the *GFP-MAP70-5* fragment was PCR amplified and cloned into the KpnI and XbaI site of the pColdTEV expression vector[35].

To generate promoter:GUS reporter lines, the promoter region was amplified with Phusion Taq Polymerase (Thermo Scientific, F531S) and cloned into the Gateway entry vector pDONR207 using BP clonase (Invitrogen, 11789-020). Transcriptional reporter constructs were made by cloning the promoter into the pKGWFS7 vector harboring the GUS reporter gene using LR clonase (Invitrogen, 11791-100).

To generate *pXCP1:GFP-EB1b*, the sequence of *GFP-EB1b* of the *35 S:GFP-EB1b*[12] was PCR-amplified and cloned into the pENTR/D-TOPO vector. The 1 kbp promoter region of *XCP1* was inserted into the NotI site of the entry vector using an In-Fusion HD cloning kit. The resulting clone was recombined with the pGWB502 vector[32].

### Transformation of Nicotiana benthamiana
Transient transformation of *N. benthamiana* leaves was performed as described previously[13].

### GUS imaging
Promoter:GUS expression analyses were performed by transferring complete seedlings or about 0.5 cm of hypocotyls or base of the floral stems to 1.5 mL of GUS staining solution (Pesquet et al., 2013) containing: 1 mM 5-bromo-4-chloro-3-indolyl-β-D-glucuronide cyclohexyl-ammonium (X-Gluc, Duchefa Biochemie, X1405.1000) dissolved in dimethylformamide (DMF), 50 mM sodium phosphate buffer (pH 7.2), 0.5 mM potassium ferricyanide (K3Fe(CN)6, Sigma Aldrich, P3667), and 0.5 mM potassium ferrocyanide (K4Fe(CN)6·3H2O, Sigma Aldrich, P9387). Samples were vacuum-infiltrated for 30 min and incubated in the dark on an orbital shaker (200 rpm) at 37 °C for 0.5-24 h, to visualize GUS activity. Samples were sequentially dehydrated in 25%, 35% and 50% ethanol solution to remove chlorophyll and fixed in FAA solution (5% formaldehyde, 10% acetic acid, and 50% ethanol) for 30 min. Fixative solution was removed by 2 washes with 50% ethanol and samples were stored in 70% ethanol at 4 °C. For visualization, 6 weeks old stem samples were rinced in water and embedded in 5% agarose prior to sectioning (100 μm) with a VT1000S vibratome (Leica, Sweden). Brightfield images of seedlings and sections were acquired using an Axioplan2 upright microscope (Zeiss, Sweden) equipped with an Axiocam color camera (Zeiss, Sweden).

### Protein preparation
The 6×His-GFP-MAP70-5 construct was transformed into *Escherichia coli* BL21-CodonPlus (DE3)-RILP competent cells (Agilent Technologies) and incubated at 37 °C until the $OD_{600}$ reached 0.4. Protein expression was induced with 0.2 mM isopropyl-b-D-thiogalactopyranoside following a temperature downshift from 37 °C to 15 °C. Cells were cultured overnight. Pellets of *E. coli* were collected, resuspended in His-extraction buffer [50 mM Tris-HCl (pH 7.5), 500 mM NaCl, 20 mM imidazole, and 10 mM ATP], and sonicated. The cell lysate was centrifuged at $3000 \times g$ for 2 min, and the supernatant was incubated with Ni Sepharose 6 Fast Flow resin (GE Healthcare) at 4 °C for 30 min. Recombinant proteins were eluted with 500 μl of His-elution buffer [50 mM Tris-HCl (pH 7.5), 500 mM NaCl, and 500 mM imidazole]. Imidazole was removed from the eluate using a PD Mini-Trap G-25 column (GE Healthcare) with PIPES buffer [10 mM PIPES (pH 6.9), 50 mM DTT, 5% sucrose, and 150 mM NaCl].

Tubulin was purified from porcine brain using the high-molarity PIPES procedure, as described previously[36]. Tubulin labeling with biotin (S0956, Tokyo Chemical Industry) or X-rhodamine (41969, Sigma-Aldrich), or Alexa Fluor 488 (Sigma-Aldrich)[37] was performed, as described previously[38]. Microtubules were polymerized in BRB80 [80 mM K-PIPES (pH 6.8), 1 mM MgCl2, and 1 mM EGTA] in the presence of 1 mM GTP and 5 μM taxol at 37 °C for 15 min, and then stabilized with additional taxol (50 μM) for 10–15 min. Microtubules were then pelleted at $230,000 \times g$ in a TLA-120.1 rotor at 25 °C, and the pellet was resuspended in BRB80 with 20 μM taxol. For double labeling microtubules with biotin and X-rhodamine, non-labeled tubulins, X-rhodamine-labeled tubulins, and biotinylated tubulins were mixed at a molar ratio of 20:1:1. For preparing Alexa Fluor 488-labeled microtubules, non-labeled tubulins and Alexa Fluor 488-labeled tubulins were mixed at a molar ratio of 40:1. The microtubules were then stored at room temperature and used within 2 weeks. ATTO565-labeled tubulin was prepared using ATTO565 NHS ester (ATTO-TEC GmbH) according to the standard technique[39]. The labeled tubulin was mixed with non-labeled tubulin at a ratio of 1:9 in a way that the concentration of tubulin in the solution 70 μM.

### Microtubule-binding assay
Flow chambers were assembled by attaching a glass coverslip (12548 A, Fisher) on a glass slide (FF-001, Matsunami) by means of a pair of double-stick tape aligned in parallel with ~3 mm separation. A PEG-coated flow chamber was incubated sequentially with the following reagents: 1.0 mg/ml α-casein for 2 min, 0.2 mg/ml neutravidin for 5 min, and X-rhodamine-labeled biotinylated microtubules for 8 min. The chamber was flush with buffer after each step. Then, MAP70-5 (3–50 nM), which was diluted in Assay Buffer [12 mM K-PIPES (pH 7.0), 1 mM MgCl2, 1 mM EGTA, 25 mM KCl, 2 mM dithiothreitol, 20 μM taxol, and oxygen scavenging mix (4.5 mg/ml glucose, 0.20 mg/ml glucose oxidase and 0.035 mg/ml catalase)] supplemented with 0.25 mg/ml α-casein, was infused into the chamber. For the microtubule-binding assay with a high concentration of MAP70-5 (50 nM), the chamber was flushed with Assay Buffer before imaging.

### Microtubule crosslinking assay
Before the assays were carried out, the glass surface was coated with a mixture of biotinylated polyethylene glycol (PEG) and nonbiotinylated PEG (Laysan Bio Inc.) to minimize nonspecific protein binding[40]. Biotinylated X-rhodamine-labeled and non-biotinylated Alexa 488-labeled microtubules, and MAP70-5 were pre-mixed in test tubes in Assay Buffer. Following a 60 min incubation at room temperature, the mixture was infused into a casein- and neutravidin-coated flow chamber and incubated for 10 min. Imaging was performed after flushing unbound microtubules with Assay Buffer.

### Microtubule one-side tethering assay
Carboxylated polystyrene microbeads (1 μm dia., 08226, Polysciences Inc.) coated with a rigor kinesin mutant (G234A)[41] were prepared as previously described[42] with slight modifications. Specifically, the rigor kinesins and microbeads were mixed at the molar ratio of ~10⁶: 1 in 0.1 M Hepes buffer (pH 8.0). The mixture was then incubated for 20 min on ice and clarified by four cycles of centrifugal spins ($9,000 \times g$, 2 min each). The rigor kinesin-coated microbeads were infused into a flow chamber and immobilized on the glass surface by nonspecific adsorption. Unbound microbeads were flushed with BRB80 supplemented with 20 μM taxol and 0.5 mg/ml α-casein. Non-biotinylated X-rhodamine-labeled microtubules were then infused into the flow chamber and incubated for 8 min. After flushing unbound microtubules with BRB80 buffer [BRB80, 20 μM taxol, 2 mM dithiothreitol, oxygen scavenging mix (4.5 mg/ml glucose, 0.20 mg/ml glucose oxidase and 0.035 mg/ml catalase), and 0.5 mg/ml α-casein], the chamber was finally filled either with 1 μM GFP-MAP70-5 or a control Pipes buffer diluted in the BRB80 buffer before imaging (streaming; exposure: 100 ms).

Microtubule persistence length was calculated using ImageJ plugins 'Kappa – Curvature Analysis' (curve input options: B-Spline mode, Open, thickness of 1 pixel; curve fitting parameters: Point Distance Minimization, threshold radius of 3 pixel) and 'PLA' (Analysis Type: Curvature Analysis). Microtubules attached to the microbeads and not immobilized on the glass surface were selected for the measurement; short microtubules (< 4 μm) were excluded because of a low fitting accuracy. For each microtubule, the coordinate positions of polymer segments were obtained using curve fitting with "Kappa". The

coordinate data from 30 frames were loaded into 'PLA' to calculate the persistence length. Similar results were obtained from the identical dataset generated using "NeuronJ" ImageJ plugin instead of 'Kappa'.

## Microtubule gliding assay

Microtubules were prepared by polymerizing labeled tubulin in the presence of GTP by incubating $56\,\mu M$ tubulin at $37\,°C$ in a polymerization buffer (80 mM Pipes, 1 mM EGTA, 5 mM MgCl$_2$, 5% DMSO, 1 mM GTP; pH-6.8). To stabilize microtubules, the microtubules solution was diluted in Taxol buffer (80 mM Pipes, 1 mM EGTA, 1 mM MgCl$_2$, 10 μM paclitaxel and -1% DMSO; pH 6.8).

GFP-kinesin-1 was prepared using a cell-free protein expression system, as previously described[43]. A linear DNA template of GFP-kinesin-1 was prepared from an original plasmid containing the first 560 amino acid residues of human kinesin-1 (Kif5b) with a green fluorescent protein (GFP) and $6\times$ His tag at the C-terminus used in the previous work[44] through PCR using a primer set of k560GFP-Fw: CCGGGATTTAGGTGACACTATAGAA and k560GFP-Rv: CTGTTAC-CAGTGGCTGCTG. We synthesized mRNAs and proteins of GFP-kinesin-1 using the WEPRO7420 Expression kit (Cell Free Science Co., Ltd.) from the linear DNA template of the GFP-kinesin-1, following the manufacturer's instructions. Synthesized kinesins were frozen and stored in liquid nitrogen after making aliquots.

Microtubule gliding assays were performed using flow cells with dimensions of $2\times10\times0.05$ mm (width $\times$ length $\times$ height) that were assembled from $10\times10$ mm to $24\times60$ mm glass coverslips (Marinefeld) with double-sided tape as the spacer. In advance of the preparation of the flow cell, the bottom glass substrate was silanized with biotin-polyethene glycol (PEG) silane 5 kDa (Laysan Bio, Inc.), following the previous method[45]. A total of 0.1 mg/ml neutravidin in 1× HKEM (10 mM Hepes pH 7.5, 50 mM KCl, 5 mM MgCl$_2$, 1 mM EGTA) buffer (3 μl) was perfused sed into the flow cell and incubated for 5 min. 2.5 μl of 0.05 mg/ml Biotinylated anti-rabbit IgG antibody (Vector Laboratories) was added into the flow cell and incubated for 5 min. 3 μl of 0.034 mg/ml rabbit anti-GFP antibody (Bioss Inc.) was bound to anti-rabbit IgG antibody and incubated for 5 min. After above each step, the excess of unbound proteins was washed away with 1% BSA in 1xHKEM (5 μl). 3 μl of 100 nM of GFP-kinesin-1 in cell-free reaction mix was deposited on GFP antibody binding surface and incubated for 3 min. The flow cell was washed with 10 μl of TicTac buffer (10 mM Hepes pH7.5, 16 mM Pipes pH 6.8, 50 mM KCl, 5 mM MgCl$_2$, 1 mM EGTA, 10 mM DTT, 10 μM paclitaxel/DMSO, 9 mg/mL D-Glucose, 17 U/mL glucose oxidase, 220 U/mL Catalase). 3 μl of ATTO565 labeled microtubules solution at the tubulin concentration of 5 μM was added into the flow cell and incubated for 2 min. After washing the flow cell with 5 μl of TicTac buffer, 7.5 μl of ATP buffer, which is the TicTac buffer supplemented with 5 mM ATP, was introduced into the flow cell to initiate the translational motion of microtubules. To test bundle formation of microtubules in the presence of MAP70, ATP buffer supplemented with 2 μM of GFP-MAP70 in a Pipes buffer (10 mM Pipes pH6.9, 50 mM DTT, 5% sucrose, 150 mM NaCl) was added into the flow cell after capturing the movies for -10 min. All the experiments above were performed at $25\,°C$[40].

## Microscopy

Metaxylem and protoxylem cells were observed at 7-days-old seedling. Localization of microtubules and pit-localized proteins were observed under an inverted fluorescence microscope (IX83-ZDC, Olympus) fitted with a confocal unit (CSU-W1, Yokogawa), a sCMOS camera (ORCA-Fusion, Hamamatsu Photonics), an UPLANSAPO 60×W water immersion lens (NA 1.2), and laser lines set at 445, 488, and 561 nm. The exposure time was adjusted between 300 msec and 1 s. For simultaneous observation of microtubules, pit-localized proteins, and cell walls, cell walls were stained with PI diluted 2000-fold. To increase PI penetration into xylem cells, the tissue was partially crushed by lightly pinching the roots with tweezers in the PI staining solution.

The structure of metaxylem cells were observed with basic fuchsin staining in ClearSee treatment[46] under an Olympus FV3000 inverted confocal microscope equipped with an UPLANSAPO 60×W water immersion lens (NA 1.2) and 561 nm laser. FV-OSR software (Olympus) was used to obtain high-resolution images. Three-dimensional images of metaxylem cells were generated by Imaris software (Bitplane Company).

The imaging of microtubules in the microtubule gliding assays was performed with a total internal reflection fluorescence (TIRF) microscope (ReLIEF, Opto-line Inc.) equipped with a Nikon 60× Plan Apo TIRF (N.A. = 1.49) oil-immersion objective and appropriate filter sets (originally designed filter cube: 50/50 mirror (Chroma), BA606/55; Nikon-FITC filter cube: EX480/30, DM505, BA535/45). Images and movies were captured using a sCMOS camera (Zyla 5.5; Andor) and Nikon NIS elements BR.

The microtubule-binding and microtubule crosslinking assays were performed using an inverted microscope (Ti, Nikon) equipped with TIRF and Epi-fluorescence imaging optics[47]. TIRF illumination was used for imaging MAP70-5; Epi illumination was used for imaging microtubules. Image acquisition was performed using NIS-Elements software (ver 4.51.00, Nikon) with a 100× objective (1.49 NA, Nikon), fluorescence filters (chroma 59022m for Epi illuminator; chroma 49002 for TIRF), and an electron-multiplying charged-coupled device camera (iXon Ultra). Still images were acquired with the exposure time of 300 ms (for MAP70-5), 300 ms (for X-rhodamine-labeled microtubules), and 600 ms (for Alexa Fluor 488-labeled microtubules). Time-lapse images were acquired with 300 ms exposure at 600 ms intervals, unless indicated in the figure legends. The acquired images were analyzed using MetaMorph (Molecular Devices) and ImageJ software (http://rsbweb.nih.gov/ij/).

## Protein variability

MAP70 orthologous protein sequences conserved in *A. thaliana*, *Nicotiana tabacum*, *Oryza sativa*, *Selaginella moellendorffii*, *Anthoceros agrestis*, *Marchantia polymorpha*, *Physcomitrella patens*, *Chara braunii*, and *Klebsormidium nitens* were obtained from GenBank/EMBL, and multiple alignments of MAP70 protein orthologs were constructed using MEGA-X software (Clustal W algorithm)[48]. Protein variability was calculated using the ConSurf Server (https://consurf.tau.ac.il/) and the multiple alignments of MAP70 orthologous proteins.

## Microtubule persistence length

The position information of microtubules was detected by TSOAX software (Parameter Settings: Gaussian Std of 1 pixel, Alpha of 0.05, MinimumSOAC Length 50 pixels)[49], and snake files containing XY information for each extracted microtubule were output. XY datasets were copied from the snake files and used for the following calculation of persistence length. The persistence length of each microtubule was calculated by PLA plugin[50] using the XY data sets with a bootstrap value of 1000.

## Microtubule crosslinking efficiency

To examine the efficiency of microtubule-crosslinking by MAP70-5, the colocalization of Alexa Fluor 488-labeled microtubules with X-rhodamine-labeled microtubules were analyzed using images acquired from microtubule crosslinking efficiency assay. First, microtubule regions were determined by setting a threshold so that the intensity of X-rhodamine signal was in the top 3% of the histogram generated over the entire image field. Next, for each microtubule region determined, the mean intensity of Alexa Fluor 488 was calculated and plotted.

## Microtubule curvature

Microtubule curvature in *N. benthamiana* was calculated using the plug-in 'Kappa – Curvature Analysis' (curve input options: B-Spline

mode, open, thickness of 1 pixel; curve fitting parameters: point distance minimization, threshold radius of 5 pixels).

## Dependence of MAP70-5 localization on microtubule curvature

To determine the dependency of MAP70-5 localization on microtubule curvature, paired images of X-rhodamine-labeled microtubules and GFP-fused MAP70-5 were processed using the plug-in 'Kappa – Curvature Analysis' (curve input options: B-Spline mode, open, thickness of 1 pixel; curve fitting parameters: point distance minimization, threshold radius of 5 pixels). Microtubule curvature and GFP-MAP70-5 intensity values generated were averaged at each bin segment (width, 0.1 μm) placed along the filament, and then plotted in a scatter graph. Prior to the Kappa analysis, images of MAP70-5 were processed for background subtraction using Gaussian blur (radius, 50 pixels).

## Reporting summary

Further information on research design is available in the Nature Portfolio Reporting Summary linked to this article.

## Data availability

The data of this study are available within the article, the Supplementary Information files and the Source Data files that accompany this article. Source data are provided with this paper.

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

## Acknowledgements

We thank N. Chua (Rockefeller University) for the pER8 vector, U. Grossniklaus (University of Zurich) for the pMDC7 vector, F. Wolter and H. Puchta (KIT Botanical Institute) for the pDe-Cas9 vector, and T. Nakagawa (Shimane University) for the pGWB vectors. This work was supported by MEXT KAKENHI grants (19H05670 and 19H05677 to Y.O.), JSPS KAKENHI grants (21H02514, 20K21435, and 23K18126 to Y.O., JP20K15141 and JP21H05886 and JP23H04425 to D.I., 21K15128 to T.S., and 22H02590 to Y. Shimamoto), the Research Grant in the Natural Sciences of the Mitsubishi Foundation (to Y.O.), Gunnar Öquist fellowship from the Kempe foundation (to E.P.), and Vetenskapsrådet (VR) research grants 2010-4620 and 2016-04727 (to E.P.).

## Author contributions

Y.O. and T.S. designed the research; T.S. performed the experiments and analyzed the data; K.S. and D.I. performed in vitro assays; Y.O. and Y. Sugiyama contributed to the preparation of materials; H.S. and E.P. performed GUS assays; T.S., E.P., Y. Shimamoto, K.S., D.I, and Y.O. wrote the manuscript.

## Competing interests

The authors declare no competing interests.
