## [Peer Review File · Nature Communications]

REVIEWER COMMENTS

Reviewer #1 (Remarks to the Author):

Xylem vessel elements usually harbor distinct patterned secondary cell walls. These patterns are formed via the specific organization of cortical microtubule array according to the action of microtubule associated proteins (MAPs). Pitted patterns are present in the metaxylem vessel elements, in which ROP-MIDD1-Kinesin-13A complex and BDR-WAL complex have been reported to define the position and boundary of pits (2D structure). In this study, Sasaki et al. report two microtubule-associated proteins, MAP70-5 and MAP70-1 that direct the growth of pit arches, developing 3D structure of pit arches.

In this study, many experiments have been well conducted. The findings are interesting, which further our understanding of how the intracellular cytoskeleton system guides the deposition of secondary cell walls to form the specific cell wall structure in xylem vessels.

I have three major concerns that may be helpful to improve this work:

1. The authors found that MAP70-5 locates at the growing tips of arches. They also showed that MAP70-5 is essential for transverse orientation of microtubules around the pits. If these findings are authentic, the mutants deficient in MAP70-5 and MAP70-1 should display two defects: the oblique pits and abnormal pit arches. However, the authors only observed oblique pits in the mutants rather than altered pit arches, which seems not to be related to the arch tip localization of MAP70-5.

2. The authors conducted a series of heterologous expression examinations in tobacco leaves and in vitro biochemical experiments to demonstrate that MAP70-5 can promote bending and bundling of microtubules. Since arranging microtubules into ring arrays is a quite novel finding, validating that expression of MAP70-5 can mediate microtubules bending into rings in Arabidopsis plants would be crucial to solidify this conclusion.

3. Based on the in vitro single-molecule analyses, the authors found that incubation of MAP70-5 increases the flexibility of microtubules, altering the physical property of microtubules. I suggested to examine the physical property of microtubules that are incubated with and without MAP70-5 using magnetic tweezer technique, which can provide a direct and important proof for the above conclusion.

Minor points:

1. Xylem vessels are usually localized deeply inside of the Arabidopsis plants. It is not easy to observe the localization of MAP70-5 protein and microtubules dynamics in metaxylem vessels in planta. However, the description of how to in situ observe MAP70-5 proteins and microtubules in vessel pits is poor and requires further details. Please provide the technique details in Method section, for example the parameters of confocal microscope in observation; what growth conditions of Arabidopsis seedlings; how long the seedlings grew; Do the seedlings being cleared before observation; how to stain cell wall and what PI concentration was used. All those information is important and helpful for the readers to follow while they conduct similar experiments.

2. The authors generated a bunch of genetic materials used for observation of MAP70 localization and microtubule organization. Please provide the primer list for generation of these relevant constructs.

Reviewer #2 (Remarks to the Author):

Review of Sasaki et al. 2023 NCB

The manuscript details findings from mutant and in vitro studies on the role of MAP70 family proteins in the construction of pit structures in developing xylem vessels. The work includes analysis of phenotype and localization at the sub-cellular level, relationship to other known genes in the pit formation pathway, and experiments in a heterologous system and in vitro to infer structure function relations for the protein. The major conclusion from the work is that the MAP70-5 and, in an overlapping manner, MAP70-1 proteins bend cortical microtubules to confine their position in nascent pits, ultimately resulting in the correct pit orientation with the developing xylem cells.

This manuscript contains some wonderful observations and several key experiments to advance our knowledge of both xylem development and the function of cortical microtubules in patterning cellular structures. The findings dramatically extend our understanding of how plant cytoskeletal elements are used to control morphogenetic process. My reading of the manuscript and examination of the figures suggests that the individual findings have been placed into a context for which the authors do not yet have sufficient evidence to support. The specific criticisms are with respect to interpretation and less to do with the excellent quality of the data sets provided. I list the criticisms in order of their importance to the scientific value of the paper. The writing is weighted somewhat more toward description than the application of description to a hypothesis test, but overall does a good job of conveying the purpose and meaning of the experiments.

Point 1: The authors describe the property of the over-expressed MAP70-5 in the heterologous system as 'bending' the cortical microtubules. They then use in vitro studies of stabilized microtubules on kinesin-1 coated coverslips to show that the microtubules, again, are 'bending'. While these properties could be related, what is described in the in vivo and in vitro systems are much more likely to be inherently different. Plant cortical microtubules show a property of treadmilling and do so at a conservative rate between 1-2 micrometers per minute. The treadmilling is possible because the cortical microtubules are stability 'attached' to the cell cortex and do not move laterally under any known condition other than when CSI1 is abrogated. The images in the manuscript of overexpression in plant cells shows the effects of induced MAP70-5 expression on a 60-90 minute time scale. If MAP70 is not altering the treadmilling property of the microtubules to stabilize them and if MAP70 is not changing the association of the microtubules to the plasma membrane, then MAP70 is not taking the existing cortical microtubules and physically causing them to bend into the shapes presented in the images. On that time scale and physical size, MAP70 is having an effect on the direction over which the microtubules are polymerizing, but not interacting to 'bend' the microtubule, similar to observations made for overexpression of AtCLASP (Kirik et al). For example, if the microtubules were marked with a photobleached spot, the action of bending would retain the spot and the polymer would assume the curved shape. This would be unprecedented because there are, to my knowledge, no other cases of microtubules being made non-dynamic at the plant cell cortex. If, however, the spot goes away within a 60-120 sec time frame, it would suggest that the direction of polymerization is being changed by MAP70 to form curved polymers, but not that the microtubules are being bent. Either way, the observation of the curved cortical polymers is of great potential significance to the field – I just do not see evidence in the manuscript that the in vivo property is to take existing microtubules and bend them into curved shapes. The action of MAP70 on animal microtubules in vitro is also very interesting and worthy of publication – but that action is not directly comparable to what is presented in the in vivo experiments.

Point 2: I do not find figure 3 to be compelling enough evidence to say that the microtubules are 'confined' to pits. It seems clear, given the excellent technical work from this group, that getting better resolved light-microscope images of the microtubules in these xylem elements is not likely possible at this time. Yet, the inference from the data that the length of the polymer(s) is more in wild type than is observed in the mutant, a central point for this manuscript, is difficult to agree with on the available data. Specifically, the authors are claiming that it is the length of the microtubules that is different where the blurring in the images makes that extremely difficult to verify. It is clear that there is a difference between wild type and mutant. But by what criteria can it be verified that the microtubules are actually being confined in the space? Is this a single polymer or overlapping

microtubules forming a bundle?

The measurements for persistence length are not well described or documented beyond the inclusion of a software name. Moreover, the images presented that I presume were used for persistence length are very crowded with microtubules and the majority of the polymers are interconnected in some way. I believe that the authors want to use persistence length as a measure of microtubule rigidity – the flexural rigidity of the extant polymer. In that case, the measurement requires evidence that the microtubules are individual polymers rather than bundles. Furthermore, the measurement should rely on the bending induced by the motor proteins either through the sheer force of pushing the microtubule through the viscous solution or through the unbalanced force of having two motors connected to the same polymer. These circumstances induce bending related to the inherent rigidity of the polymer as a function of polymer length scale. Introducing a bundling protein leads to interactions between microtubules under test that are going to appear very different from microtubules not connected in this way and are not equivalent for comparisons or related in the same way to persistence length. It is my opinion that the comparison should be between isolated single polymers attached to motors that are not interacting with other microtubules, with and without, MAP70.

Minor points:

The statistical treatment of the angles in figure 1 (C and G) should be a test of dispersion and the authors have used an ANOVA which uses dispersion to assess the probability that the mean values are different. The question under test is not whether the mean angle is different, the question is whether the distribution of angles is different. It might be better just to use the F-test to ask if the angles measured for wild type and mutant show a different variance, not a different mean.

There is no control in the manuscript to indicate tagRFP and eYFP do not have overlapping emission under the conditions of the experiments analyzing co-localization. Hence, the ratio of MAP70 and microtubule fluorescence does not rule out bleed-through in channels.

The titled statement (line 133) “MAP70 acts independently of BDR-WAL and ROP-MIDD1 pathways” is not explicitly supported by the results in the section. It is shown that the molecules are spatially separated at the time they are being imaged. But that does not exclude the molecules from having activities on each other or over time that would constitute a functional pathway. The evidence does indicate that they are spatially distinct in their localization, but not that they act independently of each other.

Line 102 should probably be ‘transverse’ axis and not ‘traverse.’

Reviewer #3 (Remarks to the Author):

The work from Takema et al. focuses on the involvement of MAP70-5 and MAP70-1 in the formation of arches and the organization of microtubules within pits. The latter are located in metaxylem vessels that ensure water transport in plants. The authors demonstrate the crucial role of both MAP70-5 and MAP70-1 in the proper orientation of pit arches, which is a precise mechanism that ensures correct alignment of pits. Importantly, the authors demonstrate that MAP70-5 is specifically associated with microtubules at the tip of arches and that it may be able to crosslink microtubules. Proper formation of pits would reduce vulnerability to cavitation during drought conditions. This is a crucial topic in plant biology.

The manuscript and figures are of very high quality, and I did not observe any inconsistencies in the study. In Figure 1, it appears that the rescued MAP70-5/map70-5 has larger pits than the control and mutants. Is it correct? It also looks like there are many more pits in map70-5 compared with WT. Did the authors consider this phenotype?

It would be interesting to know whether pit function (and/or cell wall composition) is affected in map70-5/map70-1. Do the authors have more information?

Except if I missed it, the authors did not mention anything about the plant phenotype (growth, shape). It is unclear whether the double mutant grows like the WT.

It is interesting to show that MAP70-5 is expressed in xylem cells and interfascicular fibers. It would be valuable to hypothesize about the role of MAP70-5 in fibers.

I recommend better introducing some abbreviations/constructions. For instance, the IRX3 promoter is not described.

Reviewer #1 (Remarks to the Author):

Xylem vessel elements usually harbor distinct patterned secondary cell walls. These patterns are formed via the specific organization of cortical microtubule array according to the action of microtubule associated proteins (MAPs). Pitted patterns are present in the metaxylem vessel elements, in which ROP-MIDD1-Kinesin-13A complex and BDR-WAL complex have been reported to define the position and boundary of pits (2D structure). In this study, Sasaki et al. report two microtubule-associated proteins, MAP70-5 and MAP70-1 that direct the growth of pit arches, developing 3D structure of pit arches.

In this study, many experiments have been well conducted. The findings are interesting, which further our understanding of how the intracellular cytoskeleton system guides the deposition of secondary cell walls to form the specific cell wall structure in xylem vessels.

I have three major concerns that may be helpful to improve this work:

1. The authors found that MAP70-5 locates at the growing tips of arches. They also showed that MAP70-5 is essential for transverse orientation of microtubules around the pits. If these findings are authentic, the mutants deficient in MAP70-5 and MAP70-1 should display two defects: the oblique pits and abnormal pit arches. However, the authors only observed oblique pits in the mutants rather than altered pit arches, which seems not to be related to the arch tip localization of MAP70-5.

Answer: We agree with the importance of observing the arch structure. To compare the arch structures between the wild type and the *map70* plants, we added data of cross sections of pit arches of the wild type and *map70* plants to Fig. 1F and Line #109-115.

2. The authors conducted a series of heterologous expression examinations in tobacco leaves and in vitro biochemical experiments to demonstrate that MAP70-5 can promote bending and bundling of microtubules. Since arranging microtubules into ring arrays is a quite novel finding, validating that expression of MAP70-5 can mediate

microtubules bending into rings in Arabidopsis plants would be crucial to solidify this conclusion.

Answer: Thank you for your suggestion. We expressed *GFP-MAP70-5* ectopically in Arabidopsis. As a result, bending of microtubules marked with GFP-MAP70-5 was observed in various tissues including the epidermis of leaves and hypocotyls, and root hairs. We added the data to Supplemental Fig. 4 and Line #197-199.

3. Based on the in vitro single-molecule analyses, the authors found that incubation of MAP70-5 increases the flexibility of microtubules, altering the physical property of microtubules. I suggested to examine the physical property of microtubules that are incubated with and without MAP70-5 using magnetic tweezer technique, which can provide a direct and important proof for the above conclusion.

Answer: Thank you for your suggestions. It is really nice idea. We discussed with our collaborators to test magnetic tweezer. Unfortunately, we could not construct the experimental systems effectively enough for this purpose in our environment. Instead, we constructed another system to evaluate the effect of MAP70-5 on microtubule properties by using microtubule-tethering beads. In this assay, a kinesin-head (no motor activity) coated beads were anchored to the flow chamber. Then the one ends of microtubules were tethered to the beads with the other ends being free. The microtubules exhibited swaying due to thermal fluctuation. Then we quantified the curvature of the swaying microtubules. This allows us to determine the stiffness of microtubules by calculating the persistent length of microtubules in the presence of MAP70-5. The result showed that MAP70-5 decreases stiffness of microtubules. We added the data to Fig. 6I-K and Line #283-293.

Minor points:

1. Xylem vessels are usually localized deeply inside of the Arabidopsis plants. It is not easy to observe the localization of MAP70-5 protein and microtubules dynamics in metaxylem vessels in planta. However, the description of how to in situ observe MAP70-5 proteins and microtubules in vessel pits is poor and requires further details. Please provide the technique details in Method section, for example the parameters of confocal microscope in observation; what growth conditions of Arabidopsis seedlings; how long the seedlings grew; Do the seedlings being cleared before observation; how to stain cell wall and what PI

concentration was used. All those information is important and helpful for the readers to follow while they conduct similar experiments.

Answer: We added the details of the observation conditions to Methods (Line #522-529) as below.

Line #522-529: Metaxylem and protoxylem cells were observed at 7-days-old seedling. Localization of microtubules and pit-localized proteins were observed under an inverted fluorescence microscope (IX83-ZDC, Olympus) fitted with a confocal unit (CSU-W1, Yokogawa), a sCMOS camera (ORCA-Fusion, Hamamatsu Photonics), an UPLANSAPO 60xW water immersion lens (NA 1.2), and laser lines set at 445, 488, and 561 nm. The exposure time was adjusted between 300 msec and 1 sec. For simultaneous observation of microtubules, pit-localized proteins, and cell walls, cell walls were stained with PI diluted 2000-fold. To increase PI penetration into xylem cells, the tissue was partially crushed by lightly pinching the roots with tweezers in the PI staining solution.

2. The authors generated a bunch of genetic materials used for observation of MAP70 localization and microtubule organization. Please provide the primer list for generation of these relevant constructs.

Answer: Thanks for the note, we added the primer list to Table S1.

Reviewer #2 (Remarks to the Author):

Review of Sasaki et al. 2023 NCB

The manuscript details findings from mutant and in vitro studies on the role of MAP70 family proteins in the construction of pit structures in developing xylem vessels. The work includes analysis of phenotype and localization at the sub-cellular level, relationship to other known genes in the pit formation pathway, and experiments in a heterologous system and in vitro to infer structure function relations for the protein. The major conclusion from the work is that the MAP70-5 and, in an overlapping manner, MAP70-1 proteins bend cortical microtubules to confine their position in nascent pits, ultimately resulting in the correct pit orientation with the developing xylem cells.

This manuscript contains some wonderful observations and several key experiments to advance our knowledge of both xylem development and the

function of cortical microtubules in patterning cellular structures. The findings dramatically extend our understanding of how plant cytoskeletal elements are used to control morphogenetic process. My reading of the manuscript and examination of the figures suggests that the individual findings have been placed into a context for which the authors do not yet have sufficient evidence to support. The specific criticisms are with respect to interpretation and less to do with the excellent quality of the data sets provided. I list the criticisms in order of their importance to the scientific value of the paper. The writing is weighted somewhat more toward description than the application of description to a hypothesis test, but overall does a good job of conveying the purpose and meaning of the experiments.

Point 1: The authors describe the property of the over-expressed MAP70-5 in the heterologous system as ‘bending’ the cortical microtubules. They then use in vitro studies of stabilized microtubules on kinesin-1 coated coverslips to show that the microtubules, again, are ‘bending’. While these properties could be related, what is described in the in vivo and in vitro systems are much more likely to be inherently different. Plant cortical microtubules show a property of treadmilling and do so at a conservative rate between 1-2 micrometers per minute. The treadmilling is possible because the cortical microtubules are stability ‘attached’ to the cell cortex and do not move laterally under any known condition other than when CSI1 is abrogated. The images in the manuscript of overexpression in plant cells shows the effects of induced MAP70-5 expression on a 60-90 minute time scale. If MAP70 is not altering the treadmilling property of the microtubules to stabilize them and if MAP70 is not changing the association of the microtubules to the plasma membrane, then MAP70 is not taking the existing cortical microtubules and physically causing them to bend into the shapes presented in the images. On that time scale and physical size, MAP70 is having an effect on the direction over which the microtubules are polymerizing, but not interacting to ‘bend’ the microtubule, similar to observations made for overexpression of AtCLASP (Kirik et al). For example, if the microtubules were marked with a photobleached spot, the action of bending would retain the spot and the polymer would assume the curved

shape. This would be unprecedented because there are, to my knowledge, no other cases of microtubules being made non-dynamic at the plant cell cortex. If, however, the spot goes away within a 60-120 sec time frame, it would suggest that the direction of polymerization is being changed by MAP70 to form curved polymers, but not that the microtubules are being bent. Either way, the observation of the curved cortical polymers is of great potential significance to the field –I just do not see evidence in the manuscript that the in vivo property is to take existing microtubules and bend them into curved shapes. The action of MAP70 on animal microtubules in vitro is also very interesting and worthy of publication – but that action is not directly comparable to what is presented in the in vivo experiments.

Answer: Thank you for your suggestion for photo-bleaching. We also thank you for the information about CLASP. According to your suggestion, we performed photo-bleaching of EYFP-TUB6 in leaf epidermal cells of *Nicotiana benthamiana* after expression of *MAP70-5-tagRFP*. We bleached a narrow region of the microtubules and observed the bleached spots every 30 sec (please find the figure below). The signals at the bleached spot recovered within 60 seconds. This time period was not sufficient to observe microtubule bending, and thus we thought it was difficult to distinguish whether the microtubule bending was caused by microtubule sliding or treadmilling. One possible reason for this result is that microtubules were bundled by MAP70-5 and the microtubules within the bundle polymerized into the bleached area in 60 sec. To solve this problem, it is necessary to use novel techniques such as tubulin labeling with photo-switchable fluorescent proteins. However, at this moment, this is technically challenging and it needs extended time to achieve such an experiment.

Fig. Time series of spot-bleached microtubules in *MAP70-TagRFP* overexpressing cells.
The region enclosed by the white square was bleached.

Right panels show enlarged images of the left panel.

Point 2: I do not find figure 3 to be compelling enough evidence to say that the microtubules are ‘confined’ to pits. It seems clear, given the excellent technical work from this group, that getting better resolved light-microscope images of the microtubules in these xylem elements is not likely possible at this time. Yet, the inference from the data that the length of the polymer(s) is more in wild type than is observed in the mutant, a central point for this manuscript, is difficult to agree with on the available data. Specifically, the authors are claiming that it is the length of the microtubules that is different where the blurring in the images makes that extremely difficult to verify. It is clear that there is a difference between wild type and mutant. But by what criteria can it be verified that the microtubules are actually being confined in the space? Is this a single polymer or overlapping microtubules forming a bundle?

Answer: To clarify the behavior of microtubules in the pits, we newly observed the microtubule plus-end-tracking protein EB1 in metaxylem cells. EB1 signals moved along the edge of the pits, following the pit shape in the wild type, while in the *map70* mutant, the EB1 signal moved outwards beyond the pit edge. We think that this result supports that MAP70 has a function to confine the growth of microtubules at the pit boundary. We added the data to Fig. 3E-G and Line #170-181.

The measurements for persistence length are not well described or documented beyond the inclusion of a software name. Moreover, the images presented that I presume were used for persistence length are very crowded with microtubules and the majority of the polymers are interconnected in some way. I believe that the authors want to use persistence length as a measure of microtubule rigidity – the flexural rigidity of the extant polymer. In that case, the measurement requires evidence that the microtubules are individual polymers rather than bundles. Furthermore, the measurement should rely on the bending induced by the motor proteins either through the sheer force of pushing the microtubule through the viscous solution or through the unbalanced force of having two motors connected to the same polymer. These circumstances induce bending related to the inherent rigidity of the polymer as a

function of polymer length scale. Introducing a bundling protein leads to interactions between microtubules under test that are going to appear very different from microtubules not connected in this way and are not equivalent for comparisons or related in the same way to persistence length. It is my opinion that the comparison should be between isolated single polymers attached to motors that are not interacting with other microtubules, with and without, MAP70.

Answer: We added detailed parameter information of the software on the calculation of persistence length to the methods section (Line #562-567).

We agree with you that it is ideal that single, non-bundled microtubules should be analyzed for the evaluation of microtubule rigidity. To avoid microtubule bundling, we again performed a gliding assay with short microtubules at low concentrations (Fig. 6E). In this case we projected the time-lapse images and analyzed the persistent length of the trajectory of the short microtubules. The results showed that the addition of MAP70-5 still reduced the persistence length of gliding microtubule trajectories (Fig. 6F, G). Furthermore, the process in which single microtubules bend and formed rings was observed after the addition of MAP70-5 (Fig. 6K). Additionally, we performed a new experiment to assess the stiffness of microtubules using beads-tethered microtubules. (Fig. 6 I-K). In this assay, a kinesin-head (no motor activity) coated beads were anchored to the flow chamber. Then the one ends of microtubules were tethered to the beads with the other ends being free. The microtubules exhibited swaying due to thermal fluctuation. Then we quantified the curvature of the swaying microtubules. This allows us to determine the stiffness of microtubules by calculating the persistent length of non-bundled microtubules in the presence of MAP70-5. The result showed that MAP70-5 decreases stiffness of microtubules. We added the data to Fig. 6E-K and Line #274-295.

Minor points:

The statistical treatment of the angles in figure 1 (C and G) should be a test of dispersion and the authors have used an ANOVA which uses dispersion to assess the probability that the mean values are different. The question under test is not whether the mean angle is different, the question is whether the distribution of angles is different. It might be better just to use the F-test to ask if the angles measured for wild type and mutant show a different variance, not a different mean.

Answer: In Figure 1C and G, we would like to know the mean angle of pits, not variance of the angle of pits, because *map70-1 map70-5* mutant looked to have oblique pits compared to wild type plants

which develop transverse pits. Therefore, we think it is appropriate to evaluate pit angle by the significant difference of means using ANOVA.

There is no control in the manuscript to indicate tagRFP and eYFP do not have overlapping emission under the conditions of the experiments analyzing co-localization. Hence, the ratio of MAP70 and microtubule fluorescence does not rule out bleed-through in channels.

Answer: Plants in which TagRFP-MAP70-5 or EYFP-TUB6 was expressed alone were taken under TagRFP and EYFP-TUB6 imaging conditions. As shown in the following figure, there was no fluorescence leaks of TagRFP and EYFP into channels of each other.

The titled statement (line 133) “MAP70 acts independently of BDR-WAL and ROP-MIDD1 pathways” is not explicitly supported by the results in the section. It is shown that the molecules are spatially separated at the time they are being imaged. But that does not exclude the molecules from having activities on each other or over time that would constitute a functional pathway. The evidence does indicate that they are spatially distinct in their localization, but not that they act independently of each other.

Answer: We modified the section title (Line #140) and Line #157-158 as below.

Section title: **MAP70 is spatially separated from BDR-WAL and ROP-MIDD1**

Line #157-158: This observation indicates that MAP70-5 functions independently of the BDR1-WAL

and ROP-MIDD1 pathways.

Line 102 should probably be 'transverse' axis and not 'traverse.'

Answer: Thank you for noting it. We corrected it.

Reviewer #3 (Remarks to the Author):

The work from Takema et al. focuses on the involvement of MAP70-5 and MAP70-1 in the formation of arches and the organization of microtubules within pits. The latter are located in metaxylem vessels that ensure water transport in plants. The authors demonstrate the crucial role of both MAP70-5 and MAP70-1 in the proper orientation of pit arches, which is a precise mechanism that ensures correct alignment of pits.

Importantly, the authors demonstrate that MAP70-5 is specifically associated with microtubules at the tip of arches and that it may be able to crosslink microtubules. Proper formation of pits would reduce vulnerability to cavitation during drought conditions. This is a crucial topic in plant biology.

The manuscript and figures are of very high quality, and I did not observe any inconsistencies in the study. In Figure 1, it appears that the rescued MAP70-5/map70-5 has larger pits than the control and mutants. Is it correct? It also looks like there are many more pits in map70-5 compared with WT. Did the authors consider this phenotype?

Answer: As suggested, pits of *MAP70-5/map70-5* looks a little larger than control line, but quantification did not show significant difference in pit size between those lines (right figure). To avoid misunderstanding, we have replaced *MAP70-5/map70-5* with another image (Fig. 1B).

It would be interesting to know whether pit function (and/or cell wall composition) is affected in map70-5/map70-1. Do the authors have more information?

Answer: We do not have any data on pit function so far. We would like to examine pit function in future works.

Except if I missed it, the authors did not mention anything about the plant phenotype (growth, shape). It is unclear whether the double mutant grows like the WT.

Answer: We compared the growth of WT and *map70-1 map70-5* plants and found no significant difference. We have added the image and the quantitative results to the Supplemental Fig. 2 and Line #99-100.

It is interesting to show that MAP70-5 is expressed in xylem cells and interfascicular fibers. It would be valuable to hypothesize about the role of MAP70-5 in fibers.

Answer: *MAP70-5* is also expressed in interfascicular fibers (Supplemental Fig. 3). The *MAP70* family is also expressed in a variety of other cell types. In the future, we would like to study the generality of the microtubule regulation mechanism of *MAP70* in various tissues, including interfascicular fibers.

I recommend better introducing some abbreviations/constructions. For instance, the IRX3 promoter is not described.

Answer: Thank you for noting it. We corrected.

REVIEWER COMMENTS

Reviewer #1 (Remarks to the Author):

The revised manuscript has been greatly improved by supplying several additional experiments. All my concerns have been addressed. Now, I only have a few minor issues:

1. In line 171, the authors used EB1b to show the behavior microtubules. It would be better to give a brief explanation of this protein, which may be helpful for understanding the comet track of microtubule.

2. The legend of Fig.1f does not match with the figure.

3. The authors provided solid evidence to demonstrate that MAP70-5 possesses two activities to increase the flexibility of microtubules and promotes bundling of microtubules, facilitating the formation of rings of microtubules in vitro and in vivo. It might be better to give a little speculation about this interesting molecular interaction between microtubules and MAP70s in the discussion.

Reviewer #2 (Remarks to the Author):

Review of Sasaki et al

This is a revised version of a manuscript detailing the investigation of MAP70 proteins in formation of vascular pits using Arabidopsis and a heterologous system. The discoveries include a potential role for MAP70 proteins in patterning microtubules around pit fields with subsequent effects on pit orientation. The manuscript has been edited since the prior version and new experimental data have been included to address reviewer concerns.

Line 157-158. I have to disagree with the conclusion drawn by the authors in that they are showing that the molecular elements of the prior discovered pathway, BDR1/WAL/ROP/MIDD1 do not spatially overlap – but the conclusion that they work independently is misleading. It is entirely possible that these molecules could have any number of interactions or dependencies for their function in pit formation that are not observed under these conditions – the observation just indicates that they are not co-localizing physically around the pit boundary. This is especially relevant for regulatory action on microtubules where the activation/inactivation of proteins locally has impacts over a wider spatial domain where the polymers interact. The title of the section correctly indicates the finding, but the conclusion at the end is a significant overstatement of what can be concluded from the data.

Is the inference from the observations of microtubules and EB1b at the pit domains that the pit shape (lines 175-180) is changing where the microtubules can reside, or that the microtubules are determining where the pit boundary will be located? The evidence indicates that the absence of MAP70-5 changes the localization, but the developmental context is missing in the description (i.e., is this after the pit has formed or in the formation of the pit).

The claim made in the paragraph describing the interaction of domains with microtubules (Lines 200-) concludes that there are two 'microtubule-interacting' domains. While they do show in tobacco that both domains associate to microtubules, they do not show that the binding is direct in that data. It could be the case that one domain interacts directly with the polymer and the other associates with an existing microtubule binding protein. They show that two domains localize the protein to microtubules but do not provide evidence that either is directly interacting. Noting the history of how this has

caused substantial confusion in other model systems, the description should be very clear about the evidence.

Again, in the paragraph (line 215-) and figure D asking if the partial MAP70 clones rescued the mutant phenotype – this is a wonderful experiment but it does not provide evidence that MAP70 is bending microtubules in Arabidopsis or that the bending is required for correct pit formation. Rescue of the Arabidopsis phenotype with just the half of the protein shown to induce the circular microtubule bundles in benthi would be important to include with the absence of phenotype recovery for the parts that do not induce the circular bundling. But in either case, this is evidence in favor of the hypothesis that the curvature in benthi is related to an activity in Arabidopsis, but not conclusive evidence that this is the causal action of MAP70 related to the phenotype observed in the mutant lines. Great experiments, but the conclusion is not yet merited.

The in vitro bundling expts are nice. Is this related specifically to the domain analysis suggesting two independent microtubules binding domains.

The experiments to ask if MAP70 is altering the flexibility of microtubule polymers now has more data and a more convincing approach. The data are also now related more directly to the bundling action of the proteins which appears to supersede the change in flexibility. Given the data sets, I believe the authors are within their bounds to discuss flexure. However, I would introduce a second hypothesis for the action of the protein for consideration that is also consistent with their observations. What if MAP70 is not increasing the flexural rigidity but leading to a more stabilized curvature in the polymer over its length. The bead assay is very interesting but the wavy nature of the polymers in the movies could be a result of the microtubules 'spinning' rather than waving. The same is potentially true for the tethering on the coverglass to the motor heads. I bring this to the attention of the authors because there is no a priori reason for more flexible polymers to form rings or fit within the pit domain. This idea that the polymers are forming rings because they are more flexible is not substantively grounded in the literature. Taxol, for example, stabilizes microtubules but increases their flexibility and does not 'cause' microtubule to form rings. But a polymer that is stabilized to a curve could reinforce that action when bundled with other polymers, also more energetically stable as a curve and now bundled. If the curvature in benthi arose due to increased flexure, then one might expect curved structures of larger aspect to occur in cells where the flexure is less – and that does not normally occur on the outer cell face. It does potentially occur if one considers the curvature around the anticlinal to periclinal cell face transitions. So I agree that the authors have created data sets that are consistent with increased flexibility of microtubules with the addition of MAP70. But I would caution that this is not the only interpretation consistent with the available data.

Finally, the authors include experiments in the reply to reviewers that test the hypothesis for MAP70 trapping and bending microtubules at the pit periphery. The experiment tests the idea that the microtubules are collected and bent into the space with the action of MAP70. The results, and the EB1b probe results already discussed, indicate that at least some of the microtubules at the pit periphery are dynamic and that at least some of the microtubules are turning over on a minute time scale. The authors point out, I think, that this experiment cannot tell them if there is a captive microtubule at this position held in a non-dynamic state. That would take a different probe. But what the experiment does say is that the evidence for MAP70 preventing the microtubules from polymerizing outside of the lines (the length ratio measure in the prior section) and the idea that the MAP70 is bending a polymer and holding it in that position are more difficult to support. The finding that the microtubule(s) in that position are dynamic is fascinating and needs to be incorporated into the larger narrative for how MAP70 could be acting in that space to affect the transverse pit orientation.

In sum, I find this to be one of the most interesting studies in the area of plant cell biology to come out in some time with implications for many areas of plant and cytoskeletal work. The narrative drawn for how the MAP70 proteins may be acting does not need to be so ahead of the data sets where many

of these well taken observations may have a different interpretation after more work in the area is done. Thus is the nature of studies at the leading edge of a field.

Reviewer #3 (Remarks to the Author):

I think the authors answered the reviewers' questions correctly. Consequently, I think this work should be publish

Reviewer #1 (Remarks to the Author):

The revised manuscript has been greatly improved by supplying several additional experiments. All my concerns have been addressed. Now, I only have a few minor issues:

1. In line 171, the authors used EB1b to show the behavior microtubules.

It would be better to give a brief explanation of this protein, which may be helpful for understanding the comet track of microtubule.

Answer: We add the explanation of EB1 as below.

Line #170-171: To reveal the behavior of microtubules at pit arches, we observed the microtubule end-tracking protein EB1b, which forms comet-like streaks on growing microtubule plus-ends.

2. The legend of Fig.1f does not match with the figure.

Answer: Thanks for the note. We corrected the figure legend.

3. The authors provided solid evidence to demonstrate that MAP70-5 possesses two activities to increase the flexibility of microtubules and promotes bundling of microtubules, facilitating the formation of rings of microtubules in vitro and in vivo. It might be better to give a little speculation about this interesting molecular interaction between microtubules and MAP70s in the discussion.

Answer: Thank you for your suggestion. At the current stage, however, we do not have any idea on the interaction between microtubules and MAP70 other than what we mentioned at Line #209-210: MAP70-5 has at least two sites that can interact with microtubules directly or indirectly, one in the N-terminal half and the other one in the C-terminal half.

Reviewer #2 (Remarks to the Author):

Review of Sasaki et al

This is a revised version of a manuscript detailing the investigation of MAP70 proteins in formation of vascular pits using Arabidopsis and a heterologous system. The discoveries include a potential role for MAP70 proteins in patterning microtubules around pit fields with subsequent effects on pit orientation. The manuscript has been edited since the prior version and new experimental data have been included to address reviewer concerns.

Line 157-158. I have to disagree with the conclusion drawn by the authors in that they are showing that the molecular elements of the prior discovered pathway, BDR1/WAL/ROP/MIDD1 do not spatially overlap – but the conclusion that they work independently is misleading. It is entirely possible that these molecules could have any number of interactions or dependencies for their function in pit formation that are not observed under these conditions – the observation just indicates that they are not co-localizing physically around the pit boundary. This is especially relevant for regulatory action on microtubules where the activation/inactivation of proteins locally has impacts over a wider spatial domain where the polymers interact. The title of the section correctly indicates the finding, but the conclusion at the end is a significant overstatement of what can be concluded from the data.

Answer: We modified the conclusion of the section as below.

Line #157-158: These observations indicate that MAP70-5 is spatially separated from BDR-WAL and ROP-MIDD1.

Is the inference from the observations of microtubules and EB1b at the pit domains that the pit shape (lines 175-180) is changing where the microtubules can reside, or that the microtubules are determining where the pit boundary will be located? The evidence indicates that the absence of MAP70-5 changes the localization, but the developmental context is missing in the description (i.e., is this after the pit has formed or in the formation of the pit).

Answer: We observed EB1b in developing pits, where pit arches were growing. We could distinguish the stage based on the distinct localization of microtubules along the edge of developing pit arches as shown in Figure 2D, second and third panels from the top. We added the information as below.

Line #173-175: However, the projection of the time-lapse images highlighted the distinct trajectories of EB1b, which were most likely the microtubules at the tip of growing pit arches (Fig. 3F).

The claim made in the paragraph describing the interaction of domains with microtubules (Lines 200-) concludes that there are two ‘microtubule-interacting’ domains. While they do show in tobacco that both domains associate to microtubules, they do not show that the binding is direct in that data. It could be the case that one domain interacts directly with the polymer and the other associates with an existing microtubule binding protein. They show that two domains localize the protein to microtubules but do not provide evidence that either is directly interacting. Noting the history of how this has caused substantial confusion in other model systems, the description should be very clear about the evidence.

Answer: We agree with you that we have no evidence of direct microtubule binding for MAP70. To clarify this, we modified the sentence as below.

Line #209-210: MAP70-5 has at least two sites that can interact with microtubules directly or indirectly, one in the N-terminal half and the other one in the C-terminal half.

Again, in the paragraph (line 215-) and figure D asking if the partial MAP70 clones rescued the mutant phenotype – this is a wonderful experiment but it does not provide evidence that MAP70 is bending microtubules in Arabidopsis or that the bending is required for correct pit formation. Rescue of the Arabidopsis phenotype with just the half of the protein shown to induce the circular microtubule bundles in benthi would be important to include with the absence of phenotype recovery for the parts that do not induce the circular bundling. But in either case, this is evidence in favor of the hypothesis that the curvature in benthi is related to an activity in Arabidopsis, but not conclusive evidence that this is the causal action of MAP70 related to the phenotype observed in the mutant lines. Great experiments, but the conclusion is not yet merited.

Answer: We agree with you that our data do not provide the direct evidence showing the microtubule bending is required for proper pit formation. We modified the conclusion of the section as below

Line #221-223: These results indicate that the second coiled-coil domain of MAP70-5, which possesses the microtubule-bending activity, is not required for their localization to pits but is necessary for the regulation of pit orientation.

The in vitro bundling exps are nice. Is this related specifically to the domain analysis suggesting two independent microtubules binding domains.

Answer: We have not tested the function of the two putative microtubule-binding domains of MAP70-5 in the microtubule bundling. We would like to elucidate the detailed mechanism of microtubule bundling by MAP70-5 in future.

The experiments to ask if MAP70 is altering the flexibility of microtubule polymers now has more data and a more convincing approach.

The data are also now related more directly to the bundling action of the proteins which appears to supersede the change in flexibility. Given the data sets, I believe the authors are within their bounds to discuss flexure. However, I would introduce a second hypothesis for the action of the protein for consideration that is also consistent with their observations. What if MAP70 is not increasing the flexural rigidity but leading to a more stabilized curvature in the polymer over its length.

The bead assay is very interesting but the wavy nature of the polymers in the movies could be a result of the microtubules 'spinning' rather than waving. The same is potentially true for the tethering on the coverglass to the motor heads. I bring this to the attention of the authors because there is no a priori

reason for more flexible polymers to form rings or fit within the pit domain. This idea that the polymers are forming rings because they are more flexible is not substantively grounded in the literature. Taxol, for example, stabilizes microtubules but increases their flexibility and does not 'cause' microtubule to form rings. But a polymer that is stabilized to a curve could reinforce that action when bundled with other polymers, also more energetically stable as a curve and now bundled. If the curvature in bent microtubules arose due to increased flexure, then one might expect curved structures of larger aspect to occur in cells where the flexure is less – and that does not normally occur on the outer cell face. It does potentially occur if one considers the curvature around the antipodal to pericentral cell face transitions. So I agree that the authors have created data sets that are consistent with increased flexibility of microtubules with the addition of MAP70. But I would caution that this is not the only interpretation consistent with the available data.

Answer: Thank you for your suggestion of the second hypothesis. We could not reach such hypothesis but it is really consistent with our observation. We mentioned your second hypothesis in the revised manuscript as below.

Line#292-296: These two activities of MAP70-5 perhaps promoted microtubule bending when microtubules were propelled by the glass-anchored kinesins, eventually formed rings of microtubules, although we could not exclude the possibility that the MAP70-5 induces microtubule bending by leading a stabilized curvature in the microtubule polymer, not by increasing their flexibility.

Finally, the authors include experiments in the reply to reviewers that test the hypothesis for MAP70 trapping and bending microtubules at the pit periphery. The experiment tests the idea that the microtubules are collected and bent into the space with the action of MAP70. The results, and the EB1b probe results already discussed, indicate that at least some of the microtubules at the pit periphery are dynamic and that at least some of the microtubules are turning over on a minute time scale.

The authors point out, I think, that this experiment cannot tell them if there is a captive microtubule at this position held in a non-dynamic state. That would take a different probe. But what the experiment does say is that the evidence for MAP70 preventing the microtubules from polymerizing outside of the lines (the length ratio measure in the prior section) and the idea that the MAP70 is bending a polymer and holding it in that position are more difficult to support. The finding that the microtubule(s) in that position are dynamic is fascinating and needs to be incorporated into the larger narrative for how MAP70 could be acting in that space to affect the transverse pit orientation.

Answer: We modified the result section of EB1b and added sentences to Results and Discussion regarding the dynamic microtubule behavior in pits as below.

Line#181-182: These results suggest that at least some of microtubules in the pits are highly dynamic and that MAP70 prevents the dynamic microtubules from growing beyond the pit arches.

Line#337-339: We observed elongating microtubules around the pit boundary (Fig. 3G). Such microtubule behavior might facilitate remodeling of microtubule rings to form pit arches in transverse orientation.

In sum, I find this to be one of the most interesting studies in the area of plant cell biology to come out in some time with implications for many areas of plant and cytoskeletal work. The narrative drawn for how the MAP70 proteins may be acting does not need to be so ahead of the data sets where many of these well taken observations may have a different interpretation after more work in the area is done. Thus is the nature of studies at the leading edge of a field.

We deeply appreciate your constructive suggestions and thoughtful comments. We would like to characterize MAP70 in more detail in future works.

Reviewer #3 (Remarks to the Author):

I think the authors answered the reviewers' questions correctly.

Consequently, I think this work should be publish

Answer: Thank you for your positive response.